# A Deeper Look at Optimal Transport for Imitation Learning

## Abstract

Optimal transport (OT) tools have shown early promise for imitation learning (IL) and enable a metric-aware alignment of the expert and agent's stationary distributions. Despite encouraging results, the use of OT for IL is still ad hoc and lacks a systematic treatment, which could guide future research. To help understand these inner workings, we summarize key components of prior OT-based methods for IL. We then demonstrate an instantiation of OT-based IL that attains state-of-the-art performance on a wide range of environments featuring both continuous and discrete action spaces, as well as state and image observations. Our experimentation code is public[1].

## 1 Introduction

Learning control policies from observations is a fundamental problem in many fields, such as robotics, autonomous driving and video games. If a reward signal which describes success on the task, is available, reinforcement learning (RL) offers an effective approach, which has proven successful in a number of areas (Tesauro et al., 1995; Mnih et al., 2013; 2016; Schulman et al., 2017). However, in other settings it may be difficult to design a reward function that captures complex behaviors. For example, designing a reward function to make an agent to behave in a natural or human-like way is challenging. Instead, it may be easier to demonstrate the desired behavior and train the agent to mimic it.

Imitation learning (IL) and inverse reinforcement learning (IRL) (Ng & Russell, 2000) approach the problem of training an agent from expert demonstrations. IRL methods attempt to infer the expert's underlying reward function and then use RL to train the agent using this recovered reward function. IL methods use a variety of approaches, including supervised learning (Pomerleau, 1988), uncertainty estimation (Brantley et al., 2020; Wang et al., 2019) and adversarial approaches (Ho & Ermon, 2016; Kostrikov et al., 2019). Optimal transport tools have the potential to be well-suited to IL, as they allow one to align and compare multiple agent and expert trajectories, interpreted as discrete probability measures over the agent's observation space. Various recent works build on this intuition and propose imitation learning approaches using OT, notably Dadashi et al. (2021); Papagiannis & Li (2020); Fickinger et al. (2022).

We begin by summarizing the key components of existing OT methods for IL: the observation encoding, OT cost function, choice of OT solver, reward squashing, and treatment of signals arising from distinct demonstrations. These design choices are essential as they characterize the target behavior of the trained policy and the optimization landscape. We characterize unexpected behaviors that can positively or negatively impact recovering the expert policy. We also propose an instantiation of these that we refer to as OTIL, an OT method for IL that significantly improves upon previous methods w.r.t. sample efficiency (number of interactions with the environment), simplicity, generalizability (to pixel-based learning) and performance.

A notable feature of OTIL is that it does not require access to the expert's actions, and achieves expert performance on challenging tasks from state observations alone. Also, it extends directly to pixel-based settings by leveraging representations from the RL encoder, which can be used to encode observations in a compact latent space. As a result, OTIL is also applicable to pixel-based learning without requiring any encoder pre-training via self-supervision/reconstruction or through adversarial learning.

---

[1]Code at https://anonymous.4open.science/r/OTIL_TMLR-52CC/

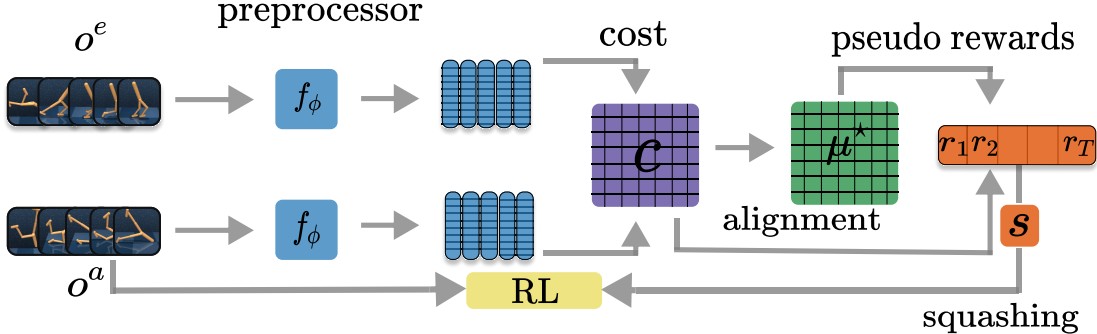

Figure 1: Visual summary of OT-based imitation learning methods. Methods $i$) encodes the agent $o^a$ and expert $o^e$ trajectories via $f_\phi$ $ii$) compute cost matrices and $iii$) an alignment matrix $\mu^\star$ between these to $iv$) produce imitation rewards $r_{1:T}$, which are squashed through $s(x)$. These rewards are then used to train agents via reinforcement learning. For clarity, we did not include the aggregation module in this illustration.

We evaluate OTIL across a range of tasks from the DeepMind Control Suite (DMC), in both the state- and pixel-based settings. We observe significant gains with respect to sample efficiency and performance compared to state-of-the-art baselines. In particular, OTIL is, to the best of our knowledge, the first approach to achieve strong performance on the quadruped benchmark from pixel observations.

## 2 Background

### 2.1 Reinforcement Learning (RL)

We instantiate RL as a discounted infinite-horizon Markov decision process (MDP) (Bellman, 1957; Sutton & Barto, 2018). In the case of pixel observations, the agent's state is approximated by a stack of consecutive RGB frames (Mnih et al., 2015). The MDP is of the form $(\mathcal{O}, \mathcal{A}, P, R, \gamma, d_0)$ where $\mathcal{O}$ and $\mathcal{A}$ are the observation and action spaces, respectively, $P : \mathcal{O} \times \mathcal{A} \to \Delta(\mathcal{O})$ is the transition function ($\Delta(\mathcal{O})$ is a distribution over $\mathcal{O}$), $R : \mathcal{O} \times \mathcal{A} \to \mathbb{R}$ is the reward function, $\gamma \in [0, 1)$ is the discount factor and $d_0$ is the initial state distribution. The RL problem consists of finding an optimal policy $\pi : \mathcal{O} \to \Delta(\mathcal{A})$ that maximizes the expected long-term reward $\mathbb{E}_\pi[\sum_{t=0}^\infty \gamma^t R(\boldsymbol{o}_t, \boldsymbol{a}_t)]$, where $\boldsymbol{o}_0 \sim d_0, \boldsymbol{a}_t \sim \pi(\cdot|\boldsymbol{o}_t)$ and $\boldsymbol{o}_{t+1} \sim P(\cdot|\boldsymbol{o}_t, \boldsymbol{a}_t)$.

### 2.2 Optimal Transport

Optimal transport (Villani, 2009; Peyré & Cuturi, 2019) tools enable comparing probability measures while incorporating the geometry of the space. The Wasserstein distance between two discrete measures $\mu_{\boldsymbol{x}} = \frac{1}{T} \sum_{t=1}^T \delta_{\boldsymbol{x}_t}$ and $\mu_{\boldsymbol{y}} = \frac{1}{T'} \sum_{t=1}^{T'} \delta_{\boldsymbol{y}_t}$ is given by

$$\mathcal{W}^2(\mu_{\boldsymbol{x}}, \mu_{\boldsymbol{y}}) = \min_{\mu \in M} \sum_{t,t'=1}^{T,T'} c(\boldsymbol{x}_t, \boldsymbol{y}_{t'})\mu_{t,t'}, \tag{1}$$

where $M = \{\mu \in \mathbb{R}^{T \times T'} : \mu \mathbf{1} = \frac{1}{T}\mathbf{1}, \mu^T\mathbf{1} = \frac{1}{T'}\mathbf{1}\}$ is the set of coupling matrices, and $c : \mathcal{O} \times \mathcal{O} \to \mathbb{R}$ is a cost function. Also, $\delta_{\boldsymbol{x}}$ refers to the Dirac measure for $\boldsymbol{x} \in \mathcal{O}$.

Intuitively, the optimal coupling $\mu^\star$ provides an alignment of samples of $\mu_{\boldsymbol{x}}$ and $\mu_{\boldsymbol{y}}$. The Wasserstein distance can hence be interpreted as the expected cost of transporting mass between aligned samples.

### 2.3 Imitation Learning via Inverse RL

In imitation learning, agents do not have access to the environment reward $R$. Instead, they are provided with a dataset of multiple expert trajectories, which the agent aims to imitate, where each trajectory is

| Agent | Preprocessor | OT Solver | Cost | Squashing | Aggregator | Extra Opt. |
|-------|--------------|-----------|------|-----------|------------|------------|
| PWIL | Fixed normalization (states) | Greedy Wasserstein | Euclidean | $\alpha e^{-\beta x}$ | Traj. concat. | No |
| SIL | Discriminator (states) | Adversarial Sinkhorn | Cosine | None | Mean | Yes |
| GWIL | None (states) | Gromov-Wasserstein | Euclidean | None | 1 demo | No |
| GDTW-IL | None (states) | Gromov-DTW | Euclidean | None | 1 demo | No |
| OTIL | Rolling norm. (states) | Sinkhorn | Cosine | None | Top-$K$ | No |
| (Sect. 3.1) | Target RL encoder (pixels) | | | | | |

Table 1: Methods for OT-based imitation learning: PWIL (Dadashi et al., 2021), SIL (Papagiannis & Li, 2020), GWIL (Fickinger et al., 2022), GDTW-IL (Cohen et al., 2021), and OTIL (Sect. 3.1).

of the form $\boldsymbol{o}^e = (\boldsymbol{o}_0^e, \ldots, \boldsymbol{o}_T^e) \in \mathcal{O}^T$. To do so, IL aims to find a policy $\pi$ so that the corresponding agent trajectories $\boldsymbol{o}^a = (\boldsymbol{o}_0^a, \ldots, \boldsymbol{o}_T^a)$ are close to expert trajectories $\boldsymbol{o}^e$ under some metric between trajectories. In recent works, OT distances introduced in Sect. 2.2 have been used to define such metrics.[2] Inverse RL is an approach to IL that designs pseudo-reward signals $r_{\text{il}}(\boldsymbol{o}_t^a)$ for agent observations $\boldsymbol{o}_t^a$, $t = 1, \ldots, T$. A policy can then be learned via RL, replacing unavailable environment rewards $R$ by pseudo-rewards $r_{\text{il}}$. For learning, the main design choices are the RL backbone used to train from rewards and a method to design rewards from observations.

## 3 Key Components of Optimal Transport-based Imitation Learning

In this section, we present the key components of recent approaches for imitation learning based on optimal transport. In this view, these approaches are composed of a common set of modules, but they differ in their design choices for each module. The modules include observation preprocessing, cost function, OT solver, pseudo-reward function, reward squashing and aggregation functions. We interpret each agent trajectory $\boldsymbol{o}^a$ as a discrete uniform probability measure with a fixed number of samples $(\boldsymbol{o}_1^a, \ldots, \boldsymbol{o}_T^a)$ living in the agent's observation space (and similarly for each expert demonstration $\boldsymbol{o}^e$). Many methods aim to design pseudo-reward signals for each observation in $\boldsymbol{o}^a$, which can then be optimized using an RL backbone.

We now describe the main components of the methods, which are illustrated in Figure 1. In Table 1, we show how recent methods including PWIL (Dadashi et al., 2021), SIL (Papagiannis & Li, 2020), GWIL (Fickinger et al., 2022), GDTW-IL (Cohen et al., 2021), along with OTIL Sect. 3.1, can be instantiated.

**Preprocessor** The preprocessor aims to extract informative state representations from observations. In the case of state-based observations, two common choices for the preprocessor function $f_\phi$ are the identity and scaling by the mean and standard deviation, so that

$$f_\phi(\boldsymbol{o}_t^a) = (\boldsymbol{o}_t^a - \boldsymbol{m}) \oslash \boldsymbol{\sigma}, \quad f_\phi(\boldsymbol{o}_t^e) = (\boldsymbol{o}_t^e - \boldsymbol{m}) \oslash \boldsymbol{\sigma}, \tag{2}$$

where $\oslash$ is the elementwise division. Here $\boldsymbol{m}, \boldsymbol{\sigma}$ are the component-wise means and standard deviations computed over a selected set of trajectories. For instance, in the case of PWIL, the statistics are computed over expert demonstrations. The preprocessor $f_\phi$ can also be a parametric function, such as a neural network. Two choices that we consider are a discriminator trained by optimizing an adversarial auxiliary loss (as in SIL), and a novel approach where we use the policy encoder itself.

**Optimal Transport Solver and Cost** The solver computes an alignment

$$\mu^\star \in \arg\min_{\mu \in M} g(\mu; f_\phi(\boldsymbol{o}^a), f_\phi(\boldsymbol{o}^e), c) \tag{3}$$

between the trajectories for an OT objective $g$ that takes the embedded agent and expert's observations as inputs and is also parameterized by a *cost function* $c$ defined in the preprocessor's output space. For example,

---

[2]Note that agent and expert trajectories are empirical proxies for the occupancy distributions under the learner and expert policies (Ho & Ermon, 2016; Dadashi et al., 2021).

the Wasserstein distance in (1) uses

$$g_{\mathcal{W}}(\mu; \boldsymbol{x}, \boldsymbol{y}, c) = \sum_{t,t'=1}^{T,T'} c(\boldsymbol{x}_t, \boldsymbol{y}_{t'}) \mu_{t,t'}, \tag{4}$$

where the cost $c$ can be the Euclidean or cosine distance, and $M$ is the set of coupling matrices. In particular, $\mathcal{W}^2(f_\phi(\boldsymbol{o}^a), f_\phi(\boldsymbol{o}^e)) = g_{\mathcal{W}}(\mu^\star; f_\phi(\boldsymbol{o}^a), f_\phi(\boldsymbol{o}^e), c)$.

Other instantiantions of (3) include Sinkhorn (Cuturi, 2013) (used in SIL), Gromov-Wasserstein (Peyré et al., 2016) (used in GWIL), GDTW (Cohen et al., 2021) (used in GDTW-IL) or CO-OT (Redko et al., 2020), which are instantiated via different choices of $g$ and $M$. Each $\mu \in M$ provides an alignment between agent and expert trajectories, i.e., intuitively, if $\mu_{t,t'} > 0$, then $\boldsymbol{o}_t^{a,\phi}$ and $\boldsymbol{o}_{t'}^{e,\phi}$ are aligned.

**Pseudo-reward Function**  The pseudo-reward function computes an intrinsic reward signal for each agent observation by comparing it to the expert observations it is aligned with. For losses relying on a linear program (e.g., Wasserstein variants or DTW variants), we can define rewards as

$$r_{\text{ot}}(\boldsymbol{o}_t^{a,\phi}) = -\sum_{t'=1}^{T,T'} C_{t,t'} \mu_{t,t'}^\star, \tag{5}$$

where $C_{t,t'} = c(f_\phi(\boldsymbol{o}_t^a), f_\phi(\boldsymbol{o}_{t'}^e))$ is the cost, and $\mu^\star$ is an alignment obtained through (3) for a choice of $g$ and $M$. In this case, rewards amount to the negative sum of costs between the agent observation and expert observations it is aligned with. For losses relying on a quadratic program (GW, GDTW,...), the pseudo-reward is of the form

$$r_{\text{ot}}(\boldsymbol{o}_t^{a,\phi}) = -\sum_{t_2,t_3,t_4=1}^{T_2,T_3,T_4} |C_{t,t_3}^a - C_{t_2,t_4}^e|^2 \mu_{t,t_3}^\star \mu_{t_2,t_4}^\star, \tag{6}$$

where $C_{t,t'}^a = c(f_\phi(\boldsymbol{o}_t^a), f_\phi(\boldsymbol{o}_{t'}^a))$, $C_{t,t'}^e = c(f_\phi(\boldsymbol{o}_t^e), f_\phi(\boldsymbol{o}_{t'}^e))$ are pairwise cost matrices. The alignment is constructed by comparing the pairwise distances between samples of each compared trajectory. Observations with similar observation neighborhood will hence be aligned, and leveraged to define rewards.

**Squashing**  The squashing function $s$ can optionally apply an exponential to the pseudo-rewards, e.g., $s(x) = \alpha e^{\beta x}$ as in PWIL. Other methods simply use the linear $s(x) = \alpha x$.

**Aggregation**  When multiple expert trajectories $\boldsymbol{o}^{e_1}, \ldots, \boldsymbol{o}^{e_N}$ are available, existing methods either combine these demonstrations by concatenating the observations and subsampling (as done in PWIL), or they combine the pseudo-rewards computed based on these demonstrations. For instance, SIL defines rewards as the mean of rewards computed from each demonstration, i.e.,

$$r_{\text{ot}}(\boldsymbol{o}_t^{a,\phi}) = \frac{1}{N} \sum_{n=1}^{N} r_{\text{ot}}^n(\boldsymbol{o}_t^{a,\phi}), \tag{7}$$

where $r_{\text{ot}}^n(\boldsymbol{o}_t^{a,\phi})$ is computed using (5) or (6) and leveraging the $n^{\text{th}}$ demonstration.

### 3.1  The OTIL instantiation

We now propose a simple and effective variant of the framework introduced in the previous section that we will refer to as Optimal Transport Imitation Learning (OTIL).

**Cost function**  Prop. 1 and Prop. 2 reveal the intricate links between the cost function $c$, the state normalization strategy $f_\phi$, and the learning dynamics. In the Euclidean case, the variance statistics of the trajectory used to standardize states act as learning rate modulators, which may differ across environments

and domains, and make the Euclidean cost a sub-optimal approach towards a method effective from states and pixels due to the scale difference. On the other hand, the cosine cost is invariant to trajectory scale, and hence to the variance of standardization. For this reason, OTIL leverages the cosine cost, i.e.,

$$C_{t,t'} = c(f_\phi(\boldsymbol{o}_t^a), f_\phi(\boldsymbol{o}_{t'}^e)) = \left[1 - \frac{\langle f_\phi(\boldsymbol{o}_t^a), f_\phi(\boldsymbol{o}_{t'}^e)\rangle}{\|f_\phi(\boldsymbol{o}_t^a)\| \, \|f_\phi(\boldsymbol{o}_{t'}^e)\|}\right] \tag{8}$$

**Preprocessing - State-based**  To preprocess state-based observations, we apply standard-scaling based on the learning agent's rollouts. Scaling statistics are updated every episode based on the current trajectory. In particular,

$$f_\phi(\boldsymbol{o}_t^a) = (\boldsymbol{o}_t^a - \boldsymbol{m}_t^a) \oslash \boldsymbol{\sigma}_t^a, \quad f_\phi(\boldsymbol{o}_t^e) = (\boldsymbol{o}_t^e - \boldsymbol{m}_t^a) \oslash \boldsymbol{\sigma}_t^a, \tag{9}$$

where $\oslash$ is the elementwise division. Here $\boldsymbol{m}_t^a, \boldsymbol{\sigma}_t^a$ are the component-wise means and standard deviations computed over the current agent trajectory.

**Preprocessing - Pixel-based**  To infer informative OT rewards, we propose to use a target network updated based on the RL encoder's weights as preprocessor for trajectories. This avoids the need to learn a network for representations in the IL part of the framework. Formally, let $\bar{\theta}$ be the slow-moving weights (updated every $P$ episode based on the current RL encoder $h_\theta$'s weights $\theta$). Then we preprocess observations in the following way:

$$f_\phi(\boldsymbol{o}_t^a) = h_{\bar{\theta}}(\boldsymbol{o}_t^a) \quad f_\phi(\boldsymbol{o}_t^e) = h_{\bar{\theta}}(\boldsymbol{o}_t^a), \tag{10}$$

**Solver**  OTIL's solver is Sinkhorn, i.e., $g$ is defined as

$$g_{\mathcal{W}}(\mu; \boldsymbol{x}, \boldsymbol{y}, c) = \sum_{t,t'=1}^{T,T'} C_{t,t'} \mu_{t,t'} - \epsilon H(\mu), \quad H(\mu) = \sum_{t,t'=1}^{T,T'} \mu_{t,t'} \log(\mu_{t,t'}) \tag{11}$$

and rewards as $r_{\text{ot}}(\boldsymbol{o}_t^{a,\phi}) = -\sum_{t'=1}^{T,T'} C_{t,t'} \mu_{t,t'}^\star$. We pick it over EMD because it is faster for long trajectories (quadratic over cubic complexity). We also pick it over DTW-based solutions because the latter are over-constrained – notably, the first time step of the agent and expert trajectories are constrained to be aligned which can be problematic in settings where the episode initialization has large variance (e.g., in Deepmind control environments).

**Aggregator**  We now discuss the choice of expert aggregation. As summarized by Proposition 3, averaging rewards can lead to an ill-defined objective that will not have any of the expert demonstrations as minimizer, and the target may be far from the shape of each trajectory. Another choice that leads to consistency is to use an argmax loss, which for each agent rollout computes rewards based on the closest expert trajectory. However, it can be problematic in the setting where all expert demonstrations are suboptimal and where it is required to aggregate signals from multiple demonstrations to mitigate suboptimality as illustrated in Figure 11 in the Acrobot environment. We hence recommend leveraging a mean over the top-$K$ closest demonstrations, which interpolates between the mean and argmax approaches. We note there is an inherent trade-off if some trajectories are low-return because the top-$K$ rewards may lead to sub-optimality if $K$ is too small as the policy targeted may hence be sub-optimal. To sum up, rewards are defined as

$$r_{\text{ot}}(\boldsymbol{o}_t^{a,\phi}) = \frac{1}{K} \sum_{k=1}^{K} r_{\text{ot}}^k(\boldsymbol{o}_t^{a,\phi}), \tag{12}$$

where $r_{\text{ot}}^k$ is the reward function that relies on the $k^{\text{th}}$ closest trajectory under the Sinkhorn loss.

**Squashing** Finally, we discuss the squashing function $s(x)$. While reward shaping with exponential squashing can improve sample efficiency by heightening the learning signals for states lying near expert trajectories, it exacerbates the impact of the evolving scale of typical observation preprocessors, which we observed led to unstability in the pixel-based case. We therefore use linear squashing $s(x) = \alpha x$.

### 3.2 Theoretical Analysis

We next theoretically analyze some of the choices for the normalization, cost, and aggregation, in order to provide insights into learning policies under OTIL's various design choices. We focus on the same-domain IL setting (agent and expert MDPs are the same) for clarity, but believe our findings will be useful for the cross-domain IL setting (agent and expert MDPs differ) in the grounding of Fickinger et al. (2022) and Cohen et al. (2021). All proofs can be found in Appendix E.

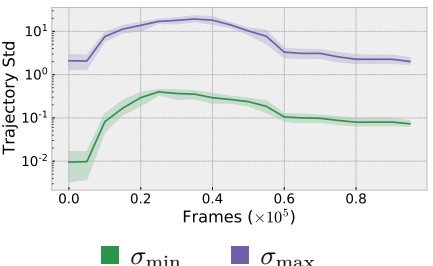

Figure 2: min and max standard deviation statistics of rollouts of an agent being trained on the cartpole swingup task. These grow significantly during exploration, and then stabilize.

**State normalization** Standard normalization of states, as described in the previous section, is commonly used in RL and IL, and has been used in an ad-hoc way in recent OT works. We study its implications on learning dynamics and the loss landscape.

**Proposition 1.** *Let* $g = g_{\mathcal{W}}$ *(see (4)), and the preprocessing strategy be standard scaling using statistics* $\boldsymbol{m}$ *and* $\boldsymbol{\sigma}$ *($f_\phi$ in (2)). If $c$ is the Euclidean cost, the sum of agent rewards computed using a single demonstration is bounded above and below by*

$$-\frac{1}{\sigma_{\min}} \sum_{t,t'=1}^{T,T'} \|(\boldsymbol{o}_t^a - \boldsymbol{o}_{t'}^e)\| \, \mu_{t,t'}^\star \leq \sum_{t=1}^{T,T'} r_{\mathrm{ot}}(\boldsymbol{o}_t^a) \leq -\frac{1}{\sigma_{\max}} \sum_{t,t'=1}^{T,T'} \|(\boldsymbol{o}_t^a - \boldsymbol{o}_{t'}^e)\| \, \mu_{t,t'}^\star. \tag{13}$$

*Proof.* If $c$ is the Euclidean cost,

$$\sum_{t=1}^{T} r_{\mathrm{ot}}(\boldsymbol{o}_t^a) = -\sum_{t,t'=1}^{T,T'} \|(\boldsymbol{o}_t^a - \boldsymbol{m} + \boldsymbol{m} - \boldsymbol{o}_{t'}^e) \oslash \boldsymbol{\sigma}\| \, \mu_{t,t'}^\star \tag{14}$$

$$= -\sum_{t,t'=1}^{T,T'} \|(\boldsymbol{o}_t^a - \boldsymbol{o}_{t'}^e) \oslash \boldsymbol{\sigma}\| \, \mu_{t,t'}^\star \tag{15}$$

$$= -\sum_{t,t'=1}^{T,T'} \sqrt{\sum_i \left[\frac{(\boldsymbol{o}_t^a)_i - (\boldsymbol{o}_{t'}^e)_i}{\sigma_i}\right]^2} \, \mu_{t,t'}^\star \tag{16}$$

$$\geq -\sum_{t,t'=1}^{T,T'} \sqrt{\sum_i \left[\frac{(\boldsymbol{o}_t^a)_i - (\boldsymbol{o}_{t'}^e)_i}{\sigma_{\min}}\right]^2} \, \mu_{t,t'}^\star \tag{17}$$

$$= -\frac{1}{\sigma_{\min}} \sum_{t,t'=1}^{T,T'} \|(\boldsymbol{o}_t^a - \boldsymbol{o}_{t'}^e)\| \, \mu_{t,t'}^\star, \tag{18}$$

where $\sigma_{\min} = \min(\sigma_1, \ldots, \sigma_d)$. The $\sigma_{\max}$ bound can be obtained equivalently. $\square$

Prop. 1 highlights a potential issue arising when optimizing the Wasserstein loss with squared Euclidean cost (this problem also extends to the Gromov–Wasserstein case). The standard deviation of the trajectory used to normalize states acts as a modulator for the reward scale and as a result for the learning rate. In Figure 2, we observe that empirically, $\sigma_{\min}$ and $\sigma_{\max}$ vary significantly during training, hence leveraging a rolling normalization (updating $\boldsymbol{m}$ and $\boldsymbol{\sigma}$ leveraging current agent rollouts) can render the optimization

landscape unstable in the Euclidean case. Fixed normalization computed using expert statistics can also be problematic in settings where the variance statistic of the expert trajectory is large, which can in practice decrease the learning rate significantly and lead to slow convergence. In contrast, for the cosine cost case,

$$\sum_{t=1}^{T} r_{\text{ot}}(\boldsymbol{o}_t^a) = -\sum_{t,t'=1}^{T,T'} \left[ 1 - \frac{\langle \boldsymbol{o}_t^a - \boldsymbol{m}, \boldsymbol{o}_{t'}^e - \boldsymbol{m} \rangle}{\|\boldsymbol{o}_t^a - \boldsymbol{m}\| \, \|\boldsymbol{o}_{t'}^e - \boldsymbol{m}\|} \right] \mu_{t,t'}^{\star}. \tag{19}$$

Equation (19) shows that the loss is independent of the variance statistic of the trajectory if $c$ is cosine, and state normalization hence amounts to mean centering. This avoids the large fluctuations in the effective learning rate (which occur when using the squared Euclidean cost), which can lead to more stable training. We note the fact of using adaptive optimizers does not mitigate for these unstable reward scales. Reward scaling is indeed essential for performance, as previously studied in Engstrom et al. (2020)

**Cost function** The choice of OT cost can also impact the landscape of solutions of the IL problem. Typical choices include the Euclidean distance (Dadashi et al., 2021) and the cosine distance (Papagiannis & Li, 2020). For simplicity, we focus our discussion on the squared Wasserstein with access to a single demonstration. We introduce the following equivalence relation:

$$\boldsymbol{o}^a \sim \boldsymbol{o}^e \text{ iff } \exists \boldsymbol{k} \in \mathbb{R}^T \text{ s.t. } \boldsymbol{o}^a = (k_1 \boldsymbol{o}_a^e, \dots, k_T \boldsymbol{o}_T^e). \tag{20}$$

**Proposition 2.** *Assume $\boldsymbol{o}^e$ and $\boldsymbol{o}^a$ have length $T$. Then the Wasserstein distance with cosine cost is a semi-metric up to scale invariance as defined in (20).*

*Proof.* A proof is provided in Appendix E $\qquad\square$

**Corollary 1.** *If $g = g_{\mathcal{W}}$ (see (4)), $c$ is the cosine cost, $T = T'$, squashing is identity, and pseudo-rewards are computed using a single expert trajectory $\boldsymbol{o}^e$, then the set of trajectories maximizing the sum of rewards $\sum_{t=1}^{T} r_{\text{ot}}(\boldsymbol{o}_t)$ consists of all elements of the scale-equivalence class introduced in (20).*

*Proof.* Given the result of Prop. 2, $\mathcal{W}$ with cosine cost between $\boldsymbol{o}_e$ and $\boldsymbol{o}_a$ is 0 iff $\boldsymbol{o}_a \sim \boldsymbol{o}_e$, i.e. if they are equivalent up to scaling. Also, it holds that

$$\sum_{t=1}^{T} r_{\text{ot}}(\boldsymbol{o}_t) = \sum_{t=1}^{T} \sum_{t'=1}^{T} \left[ 1 - \frac{\langle \boldsymbol{o}_t^a, \boldsymbol{o}_{t'}^e \rangle}{\|\boldsymbol{o}_t^a\| \, \|\boldsymbol{o}_{t'}^e\|} \right] \mu_{t,t'}^{\star} = -\mathcal{W}^2(\boldsymbol{o}^a, \boldsymbol{o}^e), \tag{21}$$

which is maximized when $\mathcal{W}^2(\boldsymbol{o}^a, \boldsymbol{o}^e) = 0$, which holds if and only if $\boldsymbol{o}^a \sim \boldsymbol{o}^e$. $\qquad\square$

Corollary 1 shows that optimizing cosine rewards amounts to aiming to recover an element of the scale-equivalence class of the expert trajectory. Invariance to scale has been shown to be a good inductive bias in other contexts (Salimans et al., 2018); since the target of the learning process is now an equivalence class, some representatives may be easier to target.

**Expert aggregation** We next study the behaviours induced by existing approaches to aggregating the signal from multiple expert demonstrations. Consider the case where we average the rewards from each expert; see (7).

**Proposition 3.** *Assume reward squashing is the identity. If $g$ is the $\mathcal{W}$ (or $\mathcal{GW}$)-induced loss, and pseudo-rewards are aggregated by averaging with (7), then the maximizer of the sum of pseudo-rewards $\sum_{t=1}^{T} r_{\text{ot}}(\boldsymbol{o}_t^a)$ is a barycenter in the measure space induced by $g$.*

*Proof.* It holds that

$$\sum_{t=1}^{T} r_{\text{ot}}(\boldsymbol{o}_t^{a,\phi}) = \sum_{t=1}^{T} \sum_{n=1}^{N} r_{\text{ot}}^n(\boldsymbol{o}_t^{a,\phi}) = -\sum_{n=1}^{N} g(\mu^{\star,n}; \boldsymbol{o}^{a,\phi}, \boldsymbol{o}^{e_n,\phi}, c). \tag{22}$$

Also, by definition, $g(\mu^{\star,n}; \boldsymbol{o}^{a,\phi}, \boldsymbol{o}^{e_n,\phi}, c) = \mathcal{W}^2(\boldsymbol{o}^{a,\phi}, \boldsymbol{o}^{e_n}, \phi)$ if $g$ induces $\mathcal{W}$ ( and $g(\mu^{\star,n}; \boldsymbol{o}^{a,\phi}, \boldsymbol{o}^{e_n,\phi}, c) = \mathcal{GW}^2(\boldsymbol{o}^{a,\phi}, \boldsymbol{o}^{e_n,\phi})$ if $g$ induces $\mathcal{GW}$). Therefore,

$$\sum_{t=1}^{T} r_{\text{ot}}(\boldsymbol{o}_t^{a,\phi}) = -\sum_{n=1}^{N} \mathcal{W}^2(\boldsymbol{o}^{a,\phi}, \boldsymbol{o}^{e_n,\phi}) \tag{23}$$

if $g$ induces $\mathcal{W}$ and similarly for $\mathcal{GW}$. The maximizer of the above objective minimizes the sum of squared Wasserstein distances (and similarly for Gromov–Wasserstein), and is therefore a barycenter.

Note this result extends to more general choices of alignments (e.g., DTW, GDTW, entropic distances..). □

Prop. 3 provides intuition for the structure of rollouts of agents trained with rewards averaged over expert demonstrations. In particular, said trajectories are barycenters under the chosen OT metric, i.e., they minimize the sum of OT distances to each individual expert trajectory. The properties of barycenters are widely studied: 2-Wasserstein barycenters are notably known to have interpolation properties, which can be problematic in imitation learning as the learned policy may be far in Wasserstein distance from each of the individual trajectories. Figure 3 illustrates this issue.

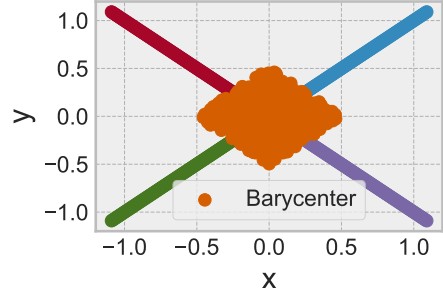

Figure 3: Barycenter of 4 linear trajectories starting at the origin and going in different directions. The barycenter does not have the shape of any of the individual trajectories.

Another typical aggregation strategy is to concatenate all trajectories and subsample them into the same shape as a single agent trajectory; see (Dadashi et al., 2021). This can also lead to undesirable policy behaviors for similar reasons as the shape of the aggregated trajectory can significantly differ from the shape of the individual trajectories.

## 4 Main Experiments

In this work, we aim to answer the following questions.

1. *What are important design choices for imitation learning approaches leveraging optimal transport?*

   OT cost – see Figure 7, state-based normalization – see Figure 8, and encoding for OT in the pixel case – see Figure 9, OT solver – see Figure 10, OT aggregation – see Figure 11, and squashing – see Figure 12.

2. *Can we design an OT approach that directly extends to pixel-based learning without extra learning?* Yes, by performing OT in the latent space of the backbone's encoder – see Figure 9 for an ablation, – see Figure 5 for main results

3. *Can non-adversarial OT methods match/improve the performance of (OT and non-OT) adversarial-based methods?* Yes, OTIL achieves equivalent or better sample efficiency and performance on most tasks from states and pixels.

4. *Does OTIL perform better than predecessor OT methods?* Yes, OTIL outperforms SIL and PWIL on all tasks from states and pixels; see Figure 4, Figure 5.

### 4.1 Experimental Setup

**Environments** We consider Mujoco (Todorov et al., 2012) tasks in the DeepMind control suite (Tassa et al., 2018), and tasks in the Arcade learning environment (Bellemare et al., 2013). The selected tasks are distinct enough to demonstrate the versatility and robustness of our approach. We experiment with state-based and pixel-based settings. We evaluate the agents with the environment rewards, but these rewards are not provided to the agents during training. Full experimental details can be found in Appendix **??**.

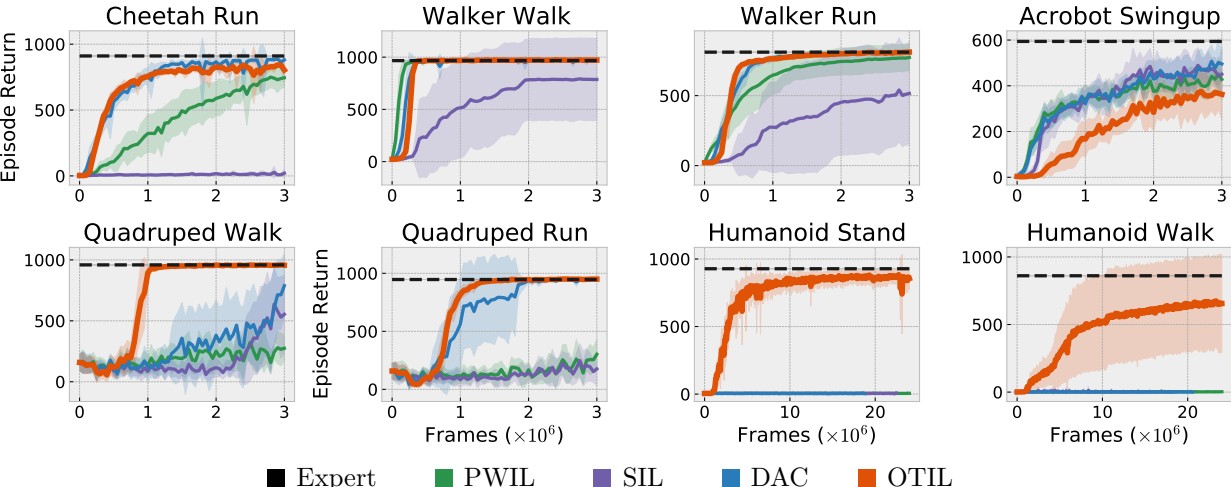

Figure 4: Episodic return of the OTIL agent along with SIL, DAC and PWIL baselines trained from state-based observations. On medium-complexity tasks (top row), all baselines achieve expert performance. On hard tasks (bottom), OTIL outperforms all baselines in terms of performance and sample efficiency by a significant margin on all tasks.

**Expert Demonstrations** For continuous control, we use DDPG with the true environment rewards. For discrete control, we use a DQN agent. We run 10 seeds and pick the seed that achieves highest episodic reward.

**Setup** In all Deepmind control experiments, we run 10 seeds under each configuration and average results with a 90% confidence interval (computed using seaborn's lineplot function) over the rewards obtained from each seed. In Atari experiments, we use 5 seeds. We compare agents with the episodic return to verify whether the agent solves the task. We provide further experimental details in App. B, and a description of the general OT for IL framework in App. C.

## 4.2 State-Based Continuous Control (DMC)

We consider two OT baselines, PWIL (Dadashi et al., 2021) and SIL (Papagiannis & Li, 2020), and a strong GAN-based baseline, DAC (Kostrikov et al., 2019). For fair comparison, we equip all baselines with the same RL backbone for learning, namely soft-actor critic (SAC) (Haarnoja et al., 2018). We train all baselines for the same number of frames and with the same noise schedule for exploration. Given that this paper focuses on the observation-only setup, behavioral cloning (Pomerleau, 1988) does not apply here as it requires expert actions. A description of SAC is provided in App. A.

In Figure 4, we observe results of all baselines on 8 representative tasks from the Deepmind control suite. On all medium-complexity tasks (first row), all methods (including ours, OTIL) but SIL perform on par. The latter has a lower sample efficiency (particularly on Cheetah Run). On harder tasks, OTIL is significantly more efficient than baselines, and is the only method to solve the hard humanoid tasks.

## 4.3 Pixel-Based Continuous Control (DMC)

For pixel-based learning we compare OTIL to SIL (Dadashi et al., 2021) and DAC (Kostrikov et al., 2019). We equip all baselines with the same SOTA pixel-based RL backbone for learning, DrQ-v2 (Yarats et al., 2021). Also, all baselines benefit from data augmentation in the form of padding, random crops and bilinear interpolation as proposed in Yarats et al. (2021). We train all baselines for the same number of frames and with the same noise schedules for exploration. A description of DrQ-v2 is provided in App. A.

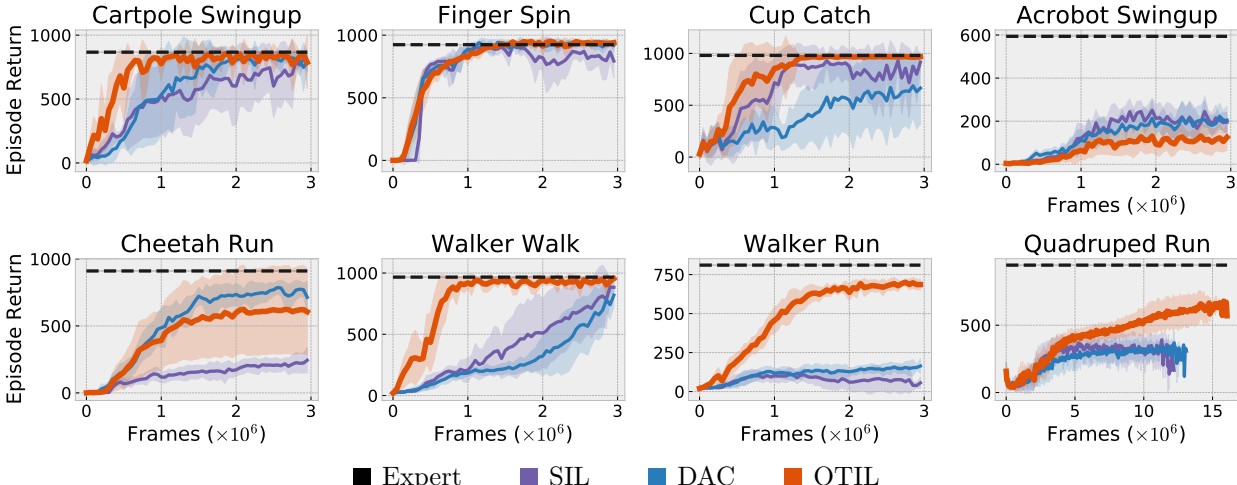

Figure 5: Episodic return of the OTIL agent along with SIL and DAC baselines trained from pixel-based observation on Deepmind control suite tasks. On medium-complexity tasks (top row), all baselines achieve expert performance. On hard tasks (bottom), OTIL outperforms baselines in terms of performance and sample efficiency by a significant margin.

We report results on 8 tasks from the control suite in Figure 5. OTIL matches or outperforms SIL and DAC on all environments besides Acrobot in terms of sample efficiency and performance.

Moreover, in contrast to the baselines, OTIL does not require training a discriminator in order to obtain representations to define rewards on, and still achieves expert performance on most tasks.

### 4.4 Pixel-Based Discrete Control (Atari)

To test the generality of our approach, we evaluate on two discrete control Atari environments, with no modification to the method besides switching to a DQN backbone (Figure 6). On these tasks, DAC and OTIL have comparable sample efficiency and performance, both matching the expert.

### 4.5 Ablations

Finally, we provide an ablation study of the main components of OTIL to study its stability to changes in hyperparameters, and report results in Figures 7-12. Note we did not re-tune all of the other hyperparameters when we ablated OTIL, hence it could be possible for a method to work well with some of

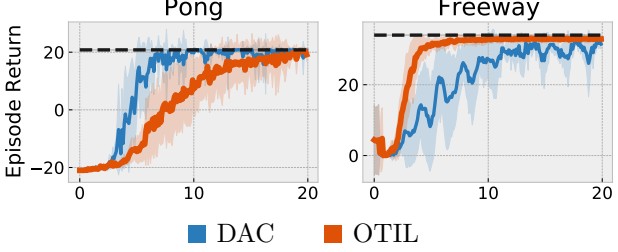

Figure 6: Episodic return of the OTIL agent and DAC trained from pixel-based observation on Atari tasks.

the less effective components if properly tuned. Note we provide full learning curves for these experiments in App. D.

**Cost function** Figure 7 shows that in practice, the cosine cost leads to better performance than the Euclidean cost (when all other OTIL components are kept fixed).

**Preprocessing** We consider agent-based standard-scaling (statistics are updated every $P$ episodes based on the current rollout), expert-based standard-scaling (statistics are computed on the expert demonstrations), no scaling, and adversarial training of a discriminator similarly to SIL (Papagiannis & Li, 2020). We observe in Figure 8 that preprocessing with agent-based standard-scaling performs best. For pixel-based learning, we compare our target network approach to adversarial learning of an encoder, a randomly initialized encoder,

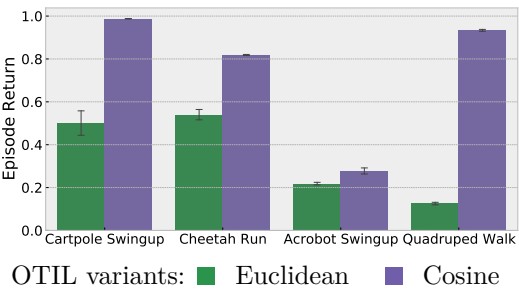

Figure 7: Episodic return (rescaled to expert return) of the OTIL agent trained with Euclidean and cosine cost. As expected by Prop. 2 and Prop. 1, the Euclidean cost leads to suboptimal performance compared to the cosine cost.

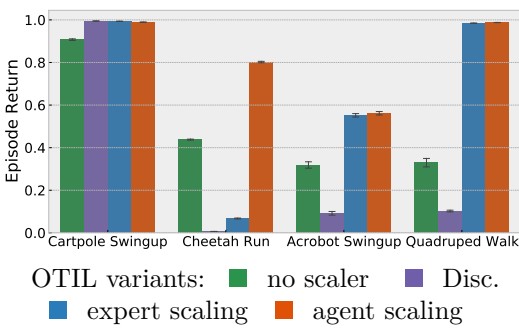

Figure 8: Episodic return (rescaled by exp.) of OTIL trained from states with different preprocessing strategies: identity, adversarially-trained discriminator, fixed normalization via expert statistics, and rolling normalization via agent statistics.

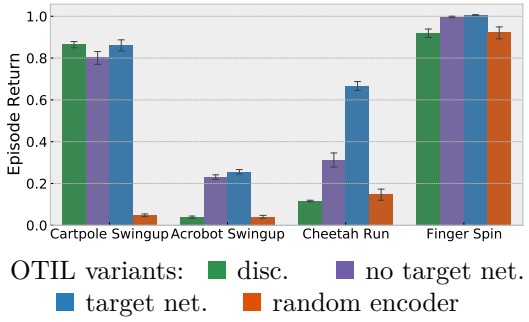

Figure 9: Episodic return (rescaled to expert return) of OTIL trained from pixels with different observation encoding strategies: adversarial training of a discriminator (disc.), encoding via RL encoder's representations (no target net.), encoding via a target network updated using the RL encoder (target net.), and finally a random encoder.

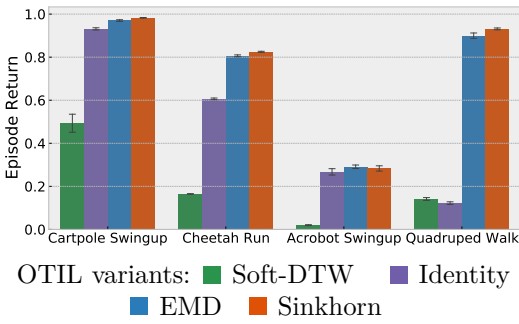

Figure 10: Episodic return (rescaled to expert return) of the OTIL agent trained from state observations with different OT losses. We consider the following strategies: Soft-DTW rewards, Identity rewards, EMD rewards (non-entropic Wasserstein) and Sinkhorn rewards (entropic Wasserstein).

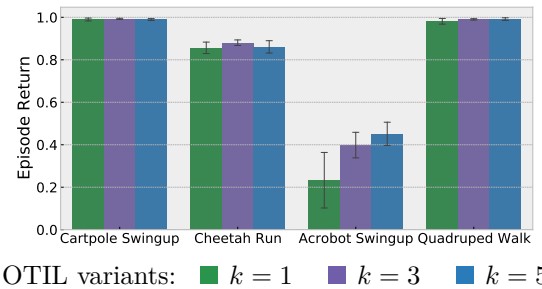

Figure 11: Episodic return (rescaled to expert return) of the OTIL agent trained from state observations with various values of $k$ in the $\text{Top} - k$ operator. For most tasks, $k = 1$ is sufficient, but on Acrobot higher ones are required to avoid converging to low-return expert trajectories, and to benefit from a more accurate estimate of the stationary expert trajectory.

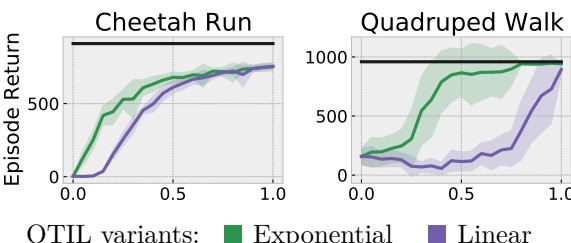

Figure 12: Episodic return of the OTIL agent trained from state observations with linear and exponential reward squashing. Exponential shaping leads to a better sample efficiency, but overall performance is equivalent.

and leveraging the RL encoder without target updates. We observe our approach performs best across all environments.

**Solver** We consider four alignment instantiations (parametrizations of $g$ and $M$): $\mathcal{W}^2$ solved via the Hungarian algorithm, $\mathcal{W}_\epsilon^2$ via Sinkhorn's algorithm, the identity, and DTW (which we approximate using a soft version named Soft-DTW (Cuturi & Blondel, 2017). As shown in Figure 10, we achieve best performance with Sinkhorn, and EMD.

**Squashing** We also report an empirical analysis of squashing functions in Figure 8. Linear and exponential squashing lead to equivalent final performance in the state-based case although the latter has better sample efficiency. However, we were not able to train any policy with exponential squashing in the pixel-based case, which we expect to be due to the significant variance of the scale of the encoder being trained. This could potentially be fixed by pre-training an encoder, and leveraging its representations to compute exponentially-squashed rewards.

## 5 Conclusion

We have given an overview of optimal transport approaches for imitation learning along with an extensive study of design choices through theory and empirical ablations. In our proposed method, OTIL, we found the choice of cost, alignment solver and observation representations essential to effective and stable learning. We demonstrated state-of-the-art performance on all tasks doing IL from states and pixels, and push the boundary of tasks solvable from pixel-based observations only.

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

## A  RL Backbones

**Soft Actor-Critic (SAC)**  For state-based continuous control, we use Soft Actor-Critic (Haarnoja et al., 2018). In SAC, we train a state-action value network $Q$, a stochastic policy network $\pi$ and a temperature $\alpha$ to solve the MDP of study. The policy is trained to maximize expected rewards while maximizing entropy.

**DrQ-v2**  For pixel-based continuous control use DrQ-v2 (Yarats et al., 2021), which is an actor-critic method based on the deep deterministic policy gradient (DDPG) algorithm (Lillicrap et al., 2015). Given a replay buffer $\mathcal{D}$, it learns simultaneously a $Q$-function $Q_\theta$ and a policy $\pi_\eta$. $Q_\theta$ is trained by clipped double-$Q$-learning (Fujimoto et al., 2018) with $n$-step returns. $\pi_\eta$ is trained via deterministic policy gradient. DrQv2 employs data augmentation in the form of random shifts with padding and a random crop to restore the original image dimension, followed by bilinear interpolation. Data augmentation acts as regularization and reduces the variance of the $Q$ estimates. Images are embedded into the latent space via an encoder $f_\phi$ after being augmented. This encoder is trained to minimize the critic loss only.

**Deep-Q-Network (DQN)**  For pixel-based discrete control, we use the Deep Q-Network method (Mnih et al., 2013). It consists in learning a convolutional network approximating the state-action value function by minimizing the squared residual error. We also leverage $n$-step returns.

## B  Hyperparameters and further experimental details.

| Agent | Parameter | Value 1 |
|---|---|---|
| Common | Replay buffer size | All (dm, states), 150000 (dm, pixels), 1M (Atari) |
| | Learning rate | $1e^{-4}$ (dm), 0.0000625 (Atari) |
| | Discount | 0.99 |
| | $n$-step returns | 1 (dm states), 3 (dm pixels, Atari) |
| | Action repeat | 2 (dm), 4 (Atari) |
| | Seed frames | 12000 (dm), 800000 (Atari) |
| | Exploration frames | 10000 (dm) |
| | Mini-batch size | 256 (dm), 64 (Atari) |
| | Agent update frequency | 2 (dm), 4 (Atari) |
| | Critic soft-update rate | 0.01 |
| | Features dim | 50 (dm), 512 (Atari) |
| | Hidden dim | 1024 (dm), 512 (Atari) |
| | Optimizer | Adam |
| | Num demos | 10 |
| | DDPG exploration schedule | $\text{linear}(1, 0.1, 1000000)$ (states) |
| | | $\text{linear}(1, 0.1, 500000)$ (pixels) |
| OTIL | Target encoder update frequency (episodes) | 20 |
| | Reward scale factor | 10 (dm states), 200 (dm pixels), 1000 (Atari) |
| | Top-$k$ | 3 (dm states), 1 (dm pixels, Atari) |
| DAC | Gradient penalty coefficient $\lambda$ | 10 |

Table 2: List of hyperparameters.

While the above set of hyperparameters is common to all environments, we found that a buffer size of 1000000 and an exploration schedule of 1000000-length was required for the pixel-based quadruped task. Also for the state-based quadruped, we used for DAC a learning rate of $5e^{-5}$ and a batch size of 512 which improved stability. Trajectories are of length 500 for Deepmind Control environments, and of variable lenght for Atari environments due to early stopping. Finally, we note that when computing rewards based on multiple expert trajectories in Figure 8, we compute statistics using the (single) concatenated trajectory (as in PWIL).

## C   Algorithmic Framework

---
**Algorithm 1** OT-IL core. Different methods can be instantiated by changing the imitation rewarder function.

---
**Require:** Expert demonstrations $\{\mu_{\boldsymbol{o}^{e_n}}\}_{n=1}^{N}$, replay buffer $\mathcal{D}$, backbone-specific networks (e.g., for policy, critic function and encoder for DrQ-v2). For adversarial baselines, also requires a discriminator $D$.

    **for** $t \in T_{total}$ **do**
        **if** done **then**
            $r_{1:T} = \text{rewarder}_{\text{imitation}}(\text{episode})$
            Update episode with $r_{1:T}$ and add all quadruples $[\boldsymbol{o}_t, \boldsymbol{a}_t, \boldsymbol{o}_{t+1}, r_t]$ to $\mathcal{D}$.
            $\boldsymbol{o}_t = \text{env.reset}()$, done = False, episode = [ ]
        **end if**
        $\boldsymbol{a}_t \sim \pi(\cdot|\boldsymbol{o}_t) \rightarrow \boldsymbol{o}_{t+1}$, done = env.step($\boldsymbol{a}_t$), episode.append($[\boldsymbol{o}_t, \boldsymbol{a}_t, \boldsymbol{o}_{t+1}]$)
        Update backbone-specific networks, and rewarder-specific networks using $\mathcal{D}$.
    **end for**

---

## D   Extra Experimental Results

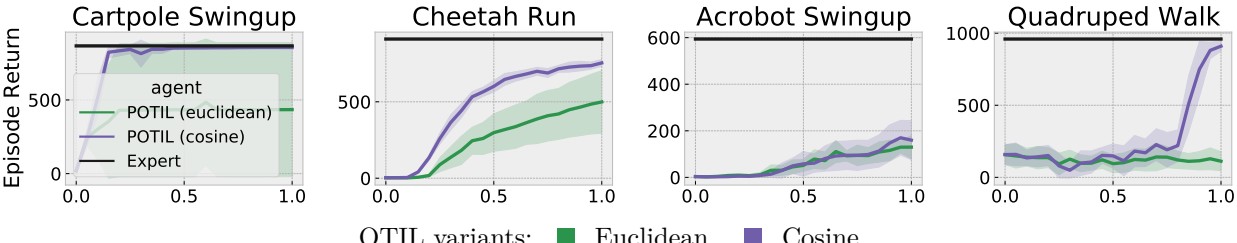

Figure 13: Episodic return of the OTIL agent trained with Euclidean and cosine cost. As expected by Prop. 2 and Prop. 1, the Euclidean cost leads to suboptimal performance compared to the cosine cost.

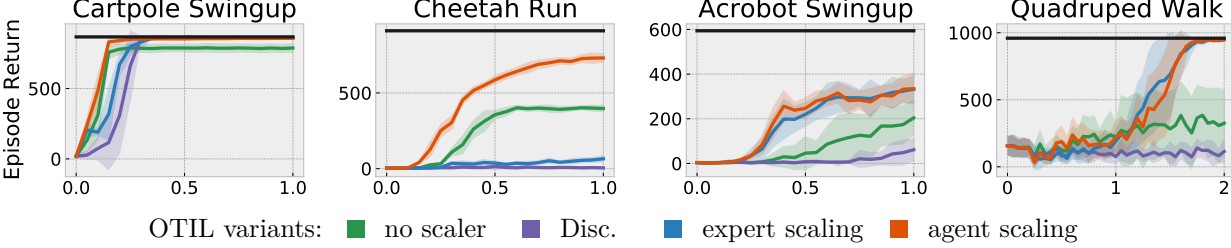

Figure 14: Episodic return of the OTIL agent trained from state observations with different observation preprocessing strategies. We consider the following strategies: no normalization, adversarial training of a discriminator, fixed normalization based on expert statistics, and nomalization based on rolling agent states.

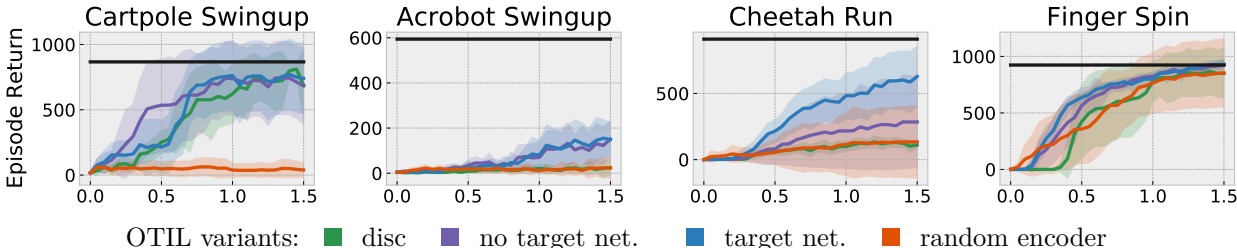

Figure 15: Episodic return of the OTIL agent trained from pixel observations with different observation encoding strategies. We consider the following strategies: adversarial training of a discriminator, encoding via RL encoder's representations, encoding via a target network updated using the RL encoder, and finally a random encoder.

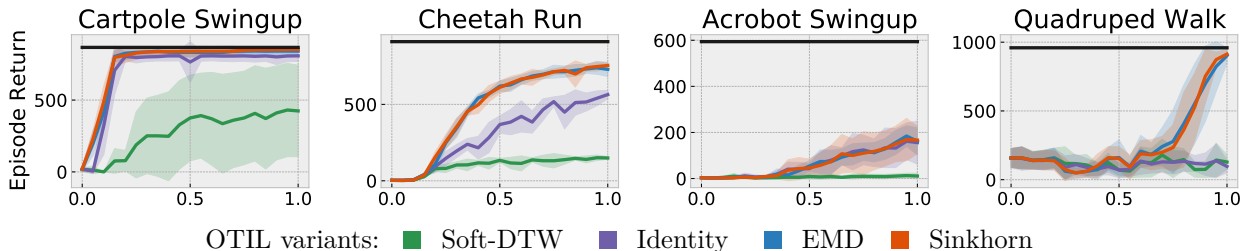

Figure 16: Episodic return of the OTIL agent trained from state observations with different OT losses. We consider the following strategies: Soft-DTW rewards, Identity rewards, EMD rewards (non-entropic Wasserstein) and Sinkhorn rewards (entropic Wasserstein).

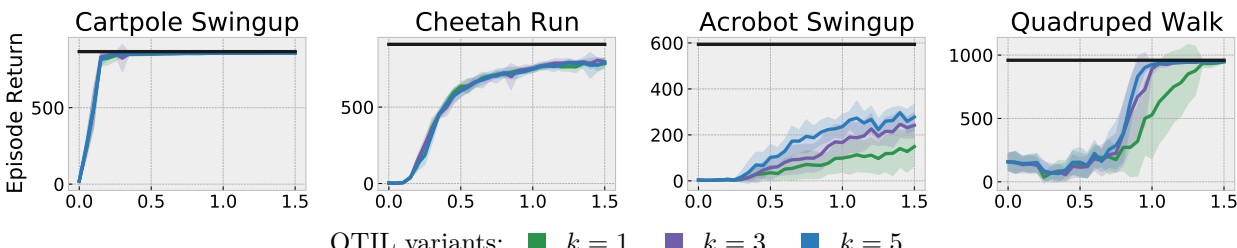

Figure 17: Episodic return of the OTIL agent trained from state observations with various values of $k$ in the Top-$k$ operator. For most tasks, $k = 1$ is sufficient, but on Acrobot higher ones are required to avoid converging to low-return expert trajectories, and to benefit from a more accurate estimate of the stationary expert trajectory.

## E   Proofs

**Proposition 4.** *Assume $\boldsymbol{o}^e$ and $\boldsymbol{o}^a$ have length $T$. Then the Wasserstein with cosine cost between $\boldsymbol{o}^e$ and $\boldsymbol{o}^a$ is a semi-metric up to scale invariance.*

*Proof.* We first show it is well-defined and positive.

$$-1 \leq \langle \frac{\boldsymbol{o}_t^a}{\|\boldsymbol{o}_t^a\|}, \frac{\boldsymbol{o}_{t'}^e}{\|\boldsymbol{o}_{t'}^e\|} \rangle \leq 1 \tag{24}$$

Hence

$$0 \leq \left[1 - \frac{\langle \boldsymbol{o}_t^a, \boldsymbol{o}_{t'}^e \rangle}{\|\boldsymbol{o}_t^a\| \|\boldsymbol{o}_{t'}^e\|}\right] \leq 2. \tag{25}$$

And as a result $\mathcal{W}$ is well defined and positive. Since $\mu_{t,t'} \geq 0$ it holds that

$$\mathcal{W}(\boldsymbol{o}^a, \boldsymbol{o}^e) = \sqrt{\sum_{t,t'=1}^{T} \left[1 - \frac{\langle \boldsymbol{o}_t^a, \boldsymbol{o}_{t'}^e \rangle}{\|\boldsymbol{o}_t^a\| \|\boldsymbol{o}_{t'}^e\|}\right] \mu_{t,t'}^{\star}} \geq 0. \tag{26}$$

We continue by showing it is symmetric.

$$\mathcal{W}(\boldsymbol{o}^a, \boldsymbol{o}^e) = \sqrt{\sum_{t,t'=1}^{T} \left[1 - \frac{\langle \boldsymbol{o}_t^a, \boldsymbol{o}_{t'}^e \rangle}{\|\boldsymbol{o}_t^a\| \|\boldsymbol{o}_{t'}^e\|}\right] \mu_{t,t'}^{\star}} = \sqrt{\sum_{t,t'=1}^{T} \left[1 - \frac{\langle \boldsymbol{o}_t^e, \boldsymbol{o}_{t'}^a \rangle}{\|\boldsymbol{o}_t^e\| \|\boldsymbol{o}_{t'}^a\|}\right] \mu_{t,t'}^{\star}} = \mathcal{W}(\boldsymbol{o}^e, \boldsymbol{o}^a). \tag{27}$$

We now discuss minimizers of $\mathcal{W}$. Define the relation $\boldsymbol{o}^1 \sim \boldsymbol{o}^2$ if and only if $\exists \boldsymbol{k} \in \mathbb{R}_+^T$, such that $\boldsymbol{o}^1 = (k_1 \boldsymbol{o}_1^2, \ldots, k_T \boldsymbol{o}_T^2)$. We show $\mathcal{W}(\boldsymbol{o}^a, \boldsymbol{o}^e) = 0$ if and only if $\boldsymbol{o}^a \sim \boldsymbol{o}^e$. Assume $\mathcal{W}(\boldsymbol{o}^a, \boldsymbol{o}^e) = 0$. By the Birkhoff–von Neumann theorem, there exists a permutation coupling induced by the permutation map $\sigma$ that is optimal for the Wasserstein with cosine cost. Hence for each $\boldsymbol{o}_t^a$, $\exists$ a single $t' = \sigma(t)$ such that $\boldsymbol{o}_t^a$ and $\boldsymbol{o}_t^e$ are aligned. The squared Wasserstein hence equals

$$\sum_{t=1}^{T} 1 - \frac{\langle \boldsymbol{o}_t^a, \boldsymbol{o}_{t'}^e \rangle}{\|\boldsymbol{o}_t^a\| \|\boldsymbol{o}_{t'}^e\|}, \tag{28}$$

which is 0 if and only if $\boldsymbol{o}_t^a = k_t \boldsymbol{o}_{t'}^e$ for all $t$, which shows the if direction.

We finally show that for any $\boldsymbol{o}^a \sim \boldsymbol{o}^e$, $\mathcal{W}(\boldsymbol{o}^a, \boldsymbol{o}^e) = 0$. Let the coupling be the identity coupling (the coupling that follows the ordering of the trajectories).

$$\sum_{t,t'}^{T} \left[1 - \frac{\langle \boldsymbol{o}_t^a, \boldsymbol{o}_{t'}^e \rangle}{\|\boldsymbol{o}_t^a\| \|\boldsymbol{o}_{t'}^e\|}\right] \mu_{t,t'}^{Id} = \sum_t^{T} \left[1 - \frac{\langle \boldsymbol{o}_t^a, \boldsymbol{o}_t^e \rangle}{\|\boldsymbol{o}_t^a\| \|\boldsymbol{o}_t^e\|}\right] = \sum_t^{T} \left[1 - \frac{\langle k_t \boldsymbol{o}_t^e, \boldsymbol{o}_t^e \rangle}{\|k_t \boldsymbol{o}_t^e\| \|\boldsymbol{o}_t^e\|}\right] = 0. \tag{29}$$

Therefore the identity coupling is optimal ($\mu^{Id} = \mu^{\star}$) and $\mathcal{W}(\boldsymbol{o}^a, \boldsymbol{o}^e) = 0$. $\qquad\square$

