# OpenReview forum: "A Deeper Look at Optimal Transport for Imitation Learning"
_TMLR — Rejected by TMLR_

### Review · Reviewer_3n8q · 2022-07-07

**Summary Of Contributions:**

The paper provides a unifying formalisation on optimal transport approaches for imitation learning. The paper studies some design choices through a theoretical analysis and ablations studies (choice of cost, alignment solver and observation representations). They also suggest their own variant (OTIL), which is shown to outperform baselines on a variety of tasks from states and pixels, and pushes the boundary of tasks solvable from pixel-based observations only.

**Broader Impact Concerns:**

No broader impact concern is required according to me.

**Requested Changes:**

- Some notations should be better defined I believe, e.g. $\delta_x$, $\delta_y$ end of page 2.
- In Fig 11, please provide the x axes labels and units.
- How are the expert trajectories provided in the experiments (how many are considered, what lengths, etc)?
- The main difference between OTIL and other algorithms is not fully clear/well highlighted. For instance, in the introduction it is mentioned that "it does not require access to the expert’s actions", however that is not only specific to OTIL.

**Strengths And Weaknesses:**

Strengths:
- The paper is well written.
- It provides both an overview (with a unifying view) as well as a new variant to optimal transport for imitation learning.
- The source code is provided

Weaknesses:
- The main differences between OTIL and other algorithms are not fully clear/well highlighted.
- Minor writing details can be improved (some notations are not defined, some label axes are not clear).

---

> ### Author Response · Authors · 2022-07-29
> **Expert trajectories**
>
> > How are the expert trajectories provided in the experiments (how many are considered, what lengths, etc)?
>
> As reported in the table of hyperparameters, we use 10 trajectories. These trajectories are of length 500 for dm_control, and of variable length for Atari environments given Atari episodes have variable length (we added these experimental details in Appendix B).

---

> ### Author Response · Authors · 2022-07-29
> **Difference to other OT works**
>
> > The main difference between OTIL and other algorithms is not fully clear/well highlighted. For instance, in the introduction it is mentioned that "it does not require access to the expert’s actions", however that is not only specific to OTIL.
>
> We provide a detailed description of differences between OTIL and other previous algorithms in Table 1. To sum up, the choice of preprocessor differs from previous works (from states we use rolling normalization using agent trajectories, and from pixels a target encoders), the choice of solver (Sinkhorn), and the choice of aggregator (top-K). The choice of cost function coincides with SIL, however SIL applies this (in the state-based case only), cost function to representations under an adversarially learned encoder while we apply it directly on the observations in the state-case, and on representations from the target network in the pixel-case.

---

> ### Author Response · Authors · 2022-07-29
> **Figure and notation**
>
> > Some notations should be better defined I believe, e.g [...]
>
> Thanks for pointing this, we improved the notation.
>
> > In Fig 11, please provide the x axes labels and units.
>
> Thanks for catching. We will update the figure to account for this.

---

> ### Comment · Reviewer_3n8q · 2022-08-10
> **My main concerns have been overall correctly addressed**
>
> Thanks to the authors for the replies. My main concern have been overall correctly addressed and explained.
>
> Some minor elements could be improved further (additional care in motivating the sensitive elements of the paper, being maybe a bit more honest about the links between the theoretical parts and the experiments as also pointed out in the other reviews). However, I believe that the paper is already a solid contribution via the useful insights and the good scientific level and I would therefore recommend acceptance.
>
> Additional suggestions:
> - When referring to a given figure or a given table followed by a number, make sure you use an uppercase (e.g. beginning Section 3 you wrote "(...) which are illustrated in fig. 1" and "In table 1, (...)").
> - it would be useful to refer more to the appendix in the main paper (I only see refs to Appendix B and E in the main paper). It is also useful to summarize in the main paper why each appendix is useful for the reader / what additional details the reader can find.

---

> > ### Author Response · Authors · 2022-08-11
> > **Thanks a lot for your feedback.**
> >
> > Thanks a lot for your time spent providing feedback. We are glad you believe the paper is a solid contribution to the field of study, and recommend acceptance.
> >
> > We fixed the hyperlinks, thanks a lot for catching this. We also added references to each individual appendices in the main paper and clarified the need for them. Also, as suggested in the other reviews, we toned down the links between the theoretical and empirical parts.

---

### Review · Reviewer_mU4T · 2022-07-14

**Summary Of Contributions:**

The paper proposes to study optimal transport for imitation learning under a unified framework. Concretely, the paper looks into the detailed empirical techniques employed by a number of existing prior work and carries out certain theoretical analysis on the properties of such design choices. These design choices include those about input preprocessing functions, OT cost functions and how rewards across multiple trajectories are aggregated. Then the paper proposes OTIL, an imitation learning algorithm based on previous theoretical insights, which have proved to be more efficient in empirical evaluations.

**Broader Impact Concerns:**

No broader impact concerns.

**Requested Changes:**

I will detail my comments and questions below. My requested changes are for the authors to address the detailed comments and questions in the rebuttal and in their revisions.

=== Sec 1.1 discussion on preprocessor

Eqn 2 details the preprocessing step. From what I understand, the normalization is carried out per trajectory? What if we have multiple trajectories do we pool the statistics m and sigma across all trajectories?

In image based control, a conv-net preprocessing module is commonly used. I think it might be better to at least give some weights to such a case in the discussion. Now it feels like most discussions are just focused on the normalization processing step, with very little mention on a learned f_phi processing function.

=== Sec 1.1 discussion on OT solver and cost

The matching plan is a T by T matrix. Does the whole OT-based IL framework implicitly assume all trajs to have the same lengths? What if trajs from expert and learner have diff lengths? Though such an assumption is valid under DM control suite, it does not hold in general.

=== Proposition 1

Prop 1 discusses how squared distance W-distance makes the resulting reward scale dependent -- i.e., depend on the scale of the observation trajectories. This motivates the cosine distance which is more scale independent. Though the argument makes sense, it feels a bit too simplistic. I would consider such a proposition and the resulting modification of the algorithm as a nice implementation "trick", instead of addressing fundamental issues in OT-based IL. One counterargument might be that current optimizers all use adaptive learning (e.g., Adam), why would a scaling in reward function be an issue?

The paper implies that when sigma is large, this makes the "learning rate" smaller. Is this issue addressed in the face of adaptive optimizer? Can we address this issue by pooling the statistics m and sigma across all trajectories in the expert dataset, such that there is just one single sigma across all learning, in which case a proper adjustment to the optimizer's learning rate would address the issue?

=== Prop 2 and Coro 1

The paper introduces a notion of equivalent class on trajectories. Is this concept useful? Equivalent class of trajectories are not necessarily feasible under the MDP / POMDP dynamics. For example, when o_t are images, o_t's pixel takes values in [0,255]. Does it make sense to scale up or down such pixel values? This will make infeasible trajectories.

Under the above argument, in the worst case, the only meaningful traj will be o^a=o^e, i.e., the learner's traj coincides with that of expert. This basically implies the learner learns perfectly. As such, the theory does not seem to be very informative, because characterizing approximate learning and potential errors induced by the learning process should be more meaningful.

=== Prop 3

The paper suggests when aggregating all trajs, if the reward is summed or averaged, it will correspond to learning a barycenter across trajectories. This argument is very simplistic because the result is a direct application of the definition of barycenter. It is true that in practice, the learner's traj can be far from individual traj from the expert data, but is this an issue? As long as expert trajs are different, the learner's traj must be far from at least one expert demonstration. How does this inform the downstream performance should be more important.

The paper suggests using top-k trajs' barycenter. Does this address the issue of "far from expert traj"? The proposed solution does not solve the problem that the paper motivates.

=== Experiment details

Pixel-based prepreocessing -- if we use a target net, does it mean the preprocessing function changes over time, albeit at a slow rate? As one motivation from the paper, we can use a preprocessing function like this without further training. Indeed, here, using a target net bypasses computational burdens. But why is this a good thing to do? What's the impact of a changing prepreocessing function? Such questions are arguably very interesting yet left unaddressed.

Solver. Why different solver should perform differently? In theory, if a solver is run until termination, it should produce very much the same result. In this work, the transport problem is a T by T discrete matching problem and can be solved optimally. Is this because diff solvers converge to diff optimal solutions? This needs further explainining.

Motivation for a pixle-based encoder without extra learning. This is a purely empirical contribution separate from the previous theoretical discussion. Having such results makes it feel that the practical techniques are a bit detached from theory, at least in the pixel-based case.

=== Side question -- imitation learning with/without states

It seems that in most existing literature uses state-dependent reward instead of state-action dependent. Is there a reason for this? I think the formulation would go through if one were to apply state-action dependent reward. How does it impact performance empirically?

=== Minor

The algorithm name "optimal transport imitation learning" is quite uninformative. It should ideally reflect differences from prior based on OT.



**Strengths And Weaknesses:**

=== Strengths ===

The strengths of the paper lie in that it has discussed a few existing prior work on optimal transport-based imitation learning, from a relatively unified framework. The empirical improvements of the proposed algorithm OTIL also seem to be significant on certain continuous control problem. The paper also shares code on the experiments, which makes the results more reproducible.

=== Weaknesses ===

Though the paper is framed as a theory-motivated paper which proposes an improved empirical algorithm based on the unifying theoretical framework, the theoretical results are a bit underwhelming. Concretely, a few potential issues with the theory are: (1) the theory feels mostly quite simplistic and straightforward, and does not seem to have significant impact from a formal theory perspective; (2) the design choices of the new empirical algorithm are not always informed by theory. Certain arguments are made in a more ad-hoc manner. This makes it difficult to assess how significantly the theory informs the practice.

---

> ### Author Response · Authors · 2022-07-29
> **Significance of theory**
>
> > Though the paper is framed as a theory-motivated paper which proposes an improved empirical algorithm based on the unifying theoretical framework, the theoretical results are a bit underwhelming. Concretely, a few potential issues with the theory are: (1) the theory feels mostly quite simplistic and straightforward, and does not seem to have significant impact from a formal theory perspective; (2) the design choices of the new empirical algorithm are not always informed by theory. Certain arguments are made in a more ad-hoc manner. This makes it difficult to assess how significantly the theory informs the practice.
>
> We would like to clarify that it was not our goal for it to be a theoretical paper. Instead, the paper provides insights that come with practical guidelines for better understanding the properties of OT-learned policies, and improving the performance of OT-based agents.

---

> ### Author Response · Authors · 2022-07-29
> **Multiple trajectories**
>
> > Eqn 2 details the preprocessing step. From what I understand, the normalization is carried out per trajectory? What if we have multiple trajectories do we pool the statistics m and sigma across all trajectories?
>
> For our approach, normalization is carried out on the statistics from the current agent trajectory. Hence we do not require pooling statistics. For PWIL, the trajectories are concatenated, and the statistics computed on the (single) concatenated trajectory. For Figure 4, in which we ablate expert-based normalization with OTIL, we compute the statistics on the concatenated expert trajectory too. We clarified this in appendix B.

---

> ### Author Response · Authors · 2022-07-29
> **Different-length trajectories**
>
> > The matching plan is a T by T matrix. Does the whole OT-based IL framework implicitly assume all trajs to have the same lengths? What if trajs from expert and learner have diff lengths? Though such an assumption is valid under DM control suite, it does not hold in general.
>
> Optimal transport allows handling different lengths-trajectories. We updated the equations to account for this in the paper. Please note that in Atari, trajectories do not have the same lengths.

---

> ### Author Response · Authors · 2022-07-29
> **Reward scaling**
>
> >Prop 1 discusses how squared distance W-distance makes the resulting reward scale dependent -- i.e., depend on the scale of the observation trajectories. This motivates the cosine distance which is more scale independent. Though the argument makes sense, it feels a bit too simplistic. I would consider such a proposition and the resulting modification of the algorithm as a nice implementation "trick", instead of addressing fundamental issues in OT-based IL. One counterargument might be that current optimizers all use adaptive learning (e.g., Adam), why would a scaling in reward function be an issue?
> The paper implies that when sigma is large, this makes the "learning rate" smaller. Is this issue addressed in the face of adaptive optimizer? Can we address this issue by pooling the statistics m and sigma across all trajectories in the expert dataset, such that there is just one single sigma across all learning, in which case a proper adjustment to the optimizer's learning rate would address the issue?
>
> Although current optimizers use adaptive scaling, reward scaling is still an important issue in modern RL, as notably pointed out in [1] (we updated the paper to refer to this). The proposed approach of pooling statistics in the expert dataset is studied in the ablation of Figure 4, and is referred to as ‘expert scaling’ (in the cosine case). We observe this significantly underperforms compared to the combined approach of rolling agent-based normalization. Please also note that although this may be due to other confounding factors, the PWIL baseline leverages mean pooling of expert statistics along with expert-based normalization (in the euclidean cost case), and also significantly underperforms compared to the OTIL baseline.
>
> [1] Implementation matters in deep policy gradients: A case study on PPO and TRPO., Engstrom et al, ICLR 2019.

---

> ### Author Response · Authors · 2022-07-29
> **Equivalence classes**
>
> >The paper introduces a notion of equivalent class on trajectories. Is this concept useful? Equivalent class of trajectories are not necessarily feasible under the MDP / POMDP dynamics. For example, when o_t are images, o_t's pixel takes values in [0,255]. Does it make sense to scale up or down such pixel values? This will make infeasible trajectories.
>
>
> Actually, we do not scale pixel values, we scale the encoded images. This is standard practice in the training of GANs with optimal transport losses [2]
>
> > Under the above argument, in the worst case, the only meaningful traj will be o^a=o^e, i.e., the learner's traj coincides with that of expert. This basically implies the learner learns perfectly. As such, the theory does not seem to be very informative, because characterizing approximate learning and potential errors induced by the learning process should be more meaningful.
>
> The fact that we work with equivalence classes means that we are aiming for our policy to target a set of equivalence classes, some of which may be easier to target than others. We provide this theory to give more insights into the use of the cosine distance in the IL setup. Empirically, leveraging the inductive biases of the cosine cost has been shown effective in the generative modeling setting, and we’ve shown these gains also extend to the IL setting. Although this does not guarantee optimality (if a member of the equivalence class is not optimal, this is typically a useful inductive bias).
>
> [2] Improving GANs using optimal transport, Salimans et al, 2018.

---

> ### Author Response · Authors · 2022-07-29
> **Barycenter of trajectories**
>
> >The paper suggests when aggregating all trajs, if the reward is summed or averaged, it will correspond to learning a barycenter across trajectories. This argument is very simplistic because the result is a direct application of the definition of barycenter. It is true that in practice, the learner's traj can be far from individual traj from the expert data, but is this an issue? As long as expert trajs are different, the learner's traj must be far from at least one expert demonstration. How does this inform the downstream performance should be more important.
> The paper suggests using top-k trajs' barycenter. Does this address the issue of "far from expert traj"? The proposed solution does not solve the problem that the paper motivates.
>
> The fact that the barycenter can be far from individual trajectories means that we may be targeting a policy that is itself far from any of the trajectories in our demonstration dataset, as the reviewer agrees. Leveraging a top-k operator over trajectories means that we will aim for our trained agent to target the barycenter of the closest trajectories in the expert dataset (to the current policy). This helps guarantee that the agent will be close to at least some of the expert trajectories (especially when K is close to 1). We note there is still a trade off because this may lead to suboptimality in the case of where some trajectories are low-return, as the policy targeted may hence be sub-optimal. We added a mention to this in the paper.

---

> ### Author Response · Authors · 2022-07-29
> **Pixel-based prepreocessing**
>
> > Pixel-based prepreocessing -- if we use a target net, does it mean the preprocessing function changes over time, albeit at a slow rate? As one motivation from the paper, we can use a preprocessing function like this without further training. Indeed, here, using a target net bypasses computational burdens. But why is this a good thing to do? What's the impact of a changing prepreocessing function? Such questions are arguably very interesting yet left unaddressed.
>
> Indeed, using a target network means that preprocessing changes across time. However, we observed the encoder converged fast to a fixed scale, and hence the function changes indeed change at a decaying rate. We provided an ablation of pixel-based preprocessing approaches in Figure 5, which shows that the target network approach performed better than without target network, than a random encoder, and than an adversarially trained encoder (which are the main typical approaches in the literature).  We agree studying these design choices further would be an interesting research direction.

---

> ### Author Response · Authors · 2022-07-29
> **Solvers**
>
> > Solver. Why different solver should perform differently? In theory, if a solver is run until termination, it should produce very much the same result. In this work, the transport problem is a T by T discrete matching problem and can be solved optimally. Is this because diff solvers converge to diff optimal solutions? This needs further explainining.
>
> Soft-DTW and identity alignments typically differ from OT solvers (Sinkhorn and EMD). Indeed, at optimality, DTW-like approaches provide alignments that have time-consistency constraints: if $t_1<t’_{1}$ and $t_2 <t’_{2}$, then it is impossible for $x_{t_1}$ to be aligned with $y_{t’_{2}}$,  and $x_{t_2}$ to be aligned with $y_{t’_{1}}$. The OT alignment is less constrained. Also, Sinkhorn and EMD alignments can also differ due to the softing coefficient of Sinkhorn, which provides smoother alignments than EMD. We clarified this in the paper.

---

> ### Author Response · Authors · 2022-07-29
> **State vs State-Action**
>
> > Side question -- imitation learning with/without states It seems that in most existing literature uses state-dependent reward instead of state-action dependent. Is there a reason for this? I think the formulation would go through if one were to apply state-action dependent reward. How does it impact performance empirically?
>
> There is a significant amount of unlabeled video data (without actions), of agents solving tasks (e.g., as studied in [3]). In order to learn from these, it is required to design methods that work in the observation-only setting. This brings us closer to scalable IL in the real world.
> [3] Playing hard exploration games by watching youtube, Aytar, Yusuf et al, NeurIPS 2018.

---

> ### Comment · Reviewer_mU4T · 2022-08-10
> **Thanks to your replies**
>
> Thanks to the authors for their replies.
>
> My questions regarding empirical aspects of the papers are mostly addressed. I have a few follow-up thoughts + questions + suggestions.
>
> #### **Significance of theory**
>
> Though the authors argue that the aim of the paper is not to be theoretical, which I agree with, my general impression after reading the theoretical part of the paper is still that it is not as closely connected to the empirical methods as I initially expected. As the authors argue, the theory is intended to guide the practice, but I do not see how the theoretical results are informative enough to guide practical designs.
>
> For example, I raised the question about the scaling argument in Prop 1 and how it motivated the cosine distance metric. Though it makes sense to think that squared losses are sensitive to the absolute scaling of the observations, it feels this is too simplistic of an argument to resort to cosine distance, provided that in practice there can be other numerical techniques that might help alleviate the issue such as optimizers based on adaptive learning rate. If cosine distance just happens to work better in practice, this is totally fine, but I do not expec the the argument in Prop 1 to completely justify that.
>
> Prop 2 uses the notion of equivalence class and semi-metric to justify the use of cosine distance metric. This feels a bit underwhelming because that the Wasserstein distance + cosine distance metric leads to a semi-metric alone, does not seem to justify the use of cosine distance either. Does it provide better geometry in the metric space? Does it entail faster optimization? Answering such questions would certainly be of interest in justifying the cosine distance metric.
>
> Overall, my concern is that though the authors might intend to emphasize less on the theory, the way the current paper is structured is that theory is claimed to play a fairly important role in guiding the practical design of OTIL. So far, I don't see how the theory is informative enough to guide the practice, either the argument feels a bit simplistic or detached from the practice. I am curious to hear what the authors have to say regarding this general concern.

---

> > ### Author Response · Authors · 2022-08-11
> > **Thanks for your useful feedback.**
> >
> > Thanks a lot for the continuous feedback, and we are glad your empirical concerns have been properly addressed!
> >
> >
> > With respect to theoretical concerns, we refactored the methodology section so the theoretical section comes after we introduce the OTIL algorithm, and frame it as an analysis section rather than a subsection used to design the OTIL algorithm. We believe the theoretical insights are still useful to understand some of OTIL’s main components, but we agree they do not fully justify the design choices (these are well justified by the empirical effectiveness demonstrated notably in Figures 4 and 5). We clarified this further at the beginning of the subsection and in the conclusion and would be happy to soften some claims even further if you believe it should be done at any specific place. Please let us know if you have any suggestions here!
> >
> > > If cosine distance just happens to work better in practice, this is totally fine, but I do not expec the the argument in Prop 1 to completely justify that.
> >
> > We agree and did not mean for Prop 1 to fully justify the use of the cosine distance, but rather to provide hints for why the Euclidean distance can be problematic in practice. We believe the refactoring of the methodology section clarifies that.
> >
> > > This feels a bit underwhelming because that the Wasserstein distance + cosine distance metric leads to a semi-metric alone, does not seem to justify the use of cosine distance either.
> >
> > Similarly, we hope the refactoring clarifies that these results are aimed to provide insights into the properties of learning policies with OT under cosine cost, but that these results are not aimed at fully justifying the use of cosine distance alone.

---

### Review · Reviewer_vqB1 · 2022-07-18

**Summary Of Contributions:**

This paper unifies several approaches for IL based on optimal transport (OT), showing how these approaches can be defined by specific design choices of (i) preprocessing function, (ii) OT solver, (iii) cost function, (iv) squashing function, and (v) aggregator. The paper then analyzes, empirically, the possible options for each one of the choices listed above, and it provides theoretical analyses for some of these options as well. Finally, the paper introduces OTIL, a new algorithm that uses what are said to be the best options for each choice and it outperforms other baselines in MuJoCO, and achieves comparable performance in Atari.



**Broader Impact Concerns:**

I have no concerns about the ethical implications of this work.



**Requested Changes:**

*Critical changes:*

- First and foremost, the paper needs to provide many more details about the empirical methodology used in the experiments about the design choices. Depending on how one sees the paper, this might seem a small thing in the big picture, but I don’t think it is, as it is listed as one of the main contributions in the Introduction and it encompasses 45% of the figures in the paper. The abstract, for example, says the paper provides an “empirical analysis of existing OT-based methods” to “gain an understanding of the[ir] inner workings”. An ablation is not that. Given all the uncertainty I discussed above, I can’t really say that the additional details would suffice for the acceptance of the paper, because it really depends on the methodology used. Nevertheless, providing the details is absolutely essential. Alternatively, the authors can soften all the claims about the role of such an empirical evaluation, clearly stating that what is provided is an ablation.

- Details on how the confidence intervals were computed need to be provided.

- The choice of using the linear squashing instead of the exponential one is justified in Figure 8 by saying that “Exponential shaping leads to a better sample efficiency, but overall performance is equivalent'', but it is said that they “found learning to be extremely unstable with the latter squashing in the pixel-based case”. Is this data available anywhere? What’s instability here? Over how many runs? I would like to see this data, since this is a general guideline that seems to be provided.

- The curves for Quadruped Run in Figure 10 are shorter for the baselines, this seems wrong.

*Minor comments [not critical]:*

- Figure 1 should probably be on page 3, with Table 1 being pushed to page 4.
- In Equation 6, what are the limits of the summation? Only the starting indices are defined as far as I can see.
- In Section 3.2, I don’t think “same-domain IL” and “cross-domain IL” are defined. I can only infer what those terms are referring to.
- \wrapfig in Figure 2 is impacting the text on page 6.
- The paper would benefit from a longer discussion around the implications of the results of Proposition 3.
- How Figure 8 is referenced in page 9 is not consistent with how the other figures are referenced in the paper.
- When using the Arcade Learning Environment, was the environment stochasticy (c.f. Machado et al., 2018)?
- The bar sizes of the norm in Equation 23 in the appendix are of different sizes.

*References*

 Machado et al: Revisiting the Arcade Learning Environment: Evaluation Protocols and Open Problems for General Agents. JAIR 2018.

**Strengths And Weaknesses:**

The paper is really well written, I enjoyed reading it and I did learn from it. I particularly liked the breakdown of the different methods in a small set of choices, which gives clarity on how these methods operate and, in a sense, grounds the discussion in the creation of new methods. The theoretical analysis is somewhat simple, but relevant to the topic being discussed and insightful (e.g., using cosine similarity instead of mean/std).

The main weakness of the paper is with respect to details about the empirical evaluation. Shortly, I don’t think there are enough details about it for me to be able to understand what was done. Specifically, for Figures 3-7, which are core to the paper as they provide the main evidence for the choices made in OTIL, how were those experiments run? Maybe I missed that, and if so, I apologize and I ask the authors to point me to the section where this is discussed. Take Figure 3, for example, which assesses the impact of using the Euclidean or Cosine distance in four different domains. What was the base algorithm used in that experiment? Is it OTIL? So, is Figure 3 showing the performance of OTIL when using the Euclidean distance vs Cosine similarity? That is, is the algorithm being evaluated here “Rolling norm. (states), Sinkhorn, {Cosine, Euclidean}, Top-K”? If that’s it, I find the analysis misleading. The discussion suggest that the choices made are better across the board (c.f. 4th paragraph of the Introduction, for example), but in fact these results would then be an ablation on OTIL. Do we expect the same conclusions to hold if the base algorithm was, say, PWIL? In fact, in some parts (e.g., Conclusion), these experiments are described as “empirical ablations”.

Obviously, I realize that doing the cross product of all possible options for each choice is infeasible, but right now, as the paper is written, the empirical analysis on the different design choices is cast as one of the main contributions. However, if OTIL was the base algorithm when generating Figures 3-7, this analysis ends up being an ablation study on OTIL. We know very well that deep RL algorithms are quite brittle and oftentimes the interaction between different components play a major role.

Another detail that is not clear to me is how parameters were swept for these experiments. Again, doing a parameter sweep before selecting the parameters for each Figure is quite expensive, but at the same time, unavoidable. How can one be confident that the parameters chosen by default are the best for the baseline? My concern is that OTIL was designed first, a good set of parameters was found, and then the alternative was tried using the same set of parameters. Obviously, in this case, what are good parameters for using, for example, the Euclidean distance, is likely different from what is a good set of parameters for using the cosine as a cost function. Clearly I’m speculating at this point and I’ll stop writing, but I wanted to highlight this because it is directly tied to my requested changes below, and I don’t think simply asking for more information about the empirical setup would suffice, depending on what it is.

The use of 10 (or 5) seeds is also an issue, because I don’t know how much can be concluded from those results, but this is pervasive in the field. Particularly, about that, how were the confidence intervals calculated in those plots? Standard normal distribution assumptions do not hold in this regime. Was this accounted for when computing the confidence intervals?

Finally, another contribution of the paper is said to be “generality (pixel-based learning)”, but almost no detail is given on the key aspects that made an OT approach that directly extends to pixel-based learning”.

*Question*

Proposition 3 of the paper assumes the identity function as preprocessor, I believe, in a sense, to provide some theoretical insight on the choice of aggregator. Proposition 3 is based on the assumption that the preprocessor is the identity function, which is not the case in the discussed algorithms; thus, is that analysis useful/informative?

Why is the question “Can non-adversarial OT (...) adversarial-based methods” interesting? I mean, the paper doesn’t make it clear why this is a meaningful question to ask.

---

> ### Author Response · Authors · 2022-07-29
> **Empirical methodology**
>
> > First and foremost, the paper needs to provide many more details about the empirical methodology used in the experiments about the design choices. Depending on how one sees the paper, this might seem a small thing in the big picture, but I don’t think it is, as it is listed as one of the main contributions in the Introduction and it encompasses 45% of the figures in the paper. The abstract, for example, says the paper provides an “empirical analysis of existing OT-based methods” to “gain an understanding of the[ir] inner workings”. An ablation is not that.  Alternatively, the authors can soften all the claims about the role of such an empirical evaluation, clearly stating that what is provided is an ablation. [...] What was the base algorithm used in that experiment? Is it OTIL?
>
> Thanks for these comments. In order to account for this, we rewrote the mentioned section in the introduction, and clarified that the experiments in Figures 3-8 are ablations of OTIL in order to soften our claims as requested.  In each experiment, the methodology consisted of taking the core OTIL algorithm and varying a single component (e.g., the cosine cost in Fig 3). We describe this at the beginning of section 4. Please note that in the original version, all captions mentioned that the base algorithm in these experiments was OTIL.

---

> ### Author Response · Authors · 2022-07-29
> **Squashing**
>
> > The choice of using the linear squashing instead of the exponential one is justified in Figure 8 by saying that “Exponential shaping leads to a better sample efficiency, but overall performance is equivalent'', but it is said that they “found learning to be extremely unstable with the latter squashing in the pixel-based case”. Is this data available anywhere? What’s instability here? Over how many runs? I would like to see this data, since this is a general guideline that seems to be provided.
>
> We did not report the result of this experiment in the pixel-based case because we were never able to successfully train pixel-based policies with exponential shaping (agents in this regime never learned any non-trivial behavior). All our experiments were always run with 10 seeds (no seed selection!), which shows this was not due to random chance. We added a remark in the paper that describes this.

---

> ### Author Response · Authors · 2022-07-29
> **Quadruped run training curves**
>
> > The curves for Quadruped Run in Figure 10 are shorter for the baselines, this seems wrong.
>
> This is explained by the fact that our local machines get jobs killed after 72 hours. Baselines are slower, hence the curves are shorter on the quadruped run. We however note that the baselines had already converged, while our approach was still learning. If requested, we can update the plot for curves to the end at 15M frames, but we would not see any further improvement of the baselines (as they had converged); it would just add more compute cost.

---

> ### Author Response · Authors · 2022-07-29
> **Minor comments**
>
> > Figure 1 should probably be on page 3, with Table 1 being pushed to page 4.
>
> Thanks for catching, we fixed this.
>
> > In Equation 6, what are the limits of the summation? Only the starting indices are defined as far as I can see.
>
> This is now fixed.
>
> > In Section 3.2, I don’t think “same-domain IL” and “cross-domain IL” are defined. I can only infer what those terms are referring to.
>
> We added a short description of those terms in the paper.
>
> > \wrapfig in Figure 2 is impacting the text on page 6.
>
> This is now fixed too.
>
> > The paper would benefit from a longer discussion around the implications of the results of Proposition 3.
>
> To update.
>
> > How Figure 8 is referenced in page 9 is not consistent with how the other figures are referenced in the paper.
>
> This is now fixed.
>
> > When using the Arcade Learning Environment, was the environment stochasticy (c.f. Machado et al., 2018)?
>
> We did not use sticky actions, hence there was no environmental stochasticity.
>
> > The bar sizes of the norm in Equation 23 in the appendix are of different sizes.
>
> We fixed this in the paper.

---

> > ### Comment · Action_Editors · 2022-08-03
> > **Clarification**
> >
> > Dear Authors,
> >
> > Thank you for your rebuttal.
> >
> > Could you please exapand on the "To update" comment in your reply?
> > For example, can you please share what you are planning to include/discuss, at least at a high level?
> >
> > Thanks

---

> > > ### Author Response · Authors · 2022-08-04
> > > **Proposition 3**
> > >
> > > Of course, thanks for putting this forward! In order to clarify the implications of Proposition 3 further, we illustrated the theoretical result with the computation of the barycenter of 4 linear trajectories starting at the origin and going in different directions in Figure 3. We observe that the barycenter is not a line, and does not lie close to any of the individual trajectories, which shows a potential failure case of average rewards in the OT for IL framework. Note the top-1 barycenter in this case would be the closest line to the agent rollout.

---

### Comment · Reviewer_vqB1 · 2022-08-03
**Comment after rebuttal**

I have just read all responses submitted by the authors as well as the changes made to the manuscript. We’re now supposed to discuss and make a final recommendation. Right now my recommendation is **reject** (I haven’t submitted it yet, I just wanted to be upfront about it). My main rationale is that my concerns were barely addressed in the rebuttal and that is quite upsetting given how much time it takes to iterate over these things. I just spent more time reading everything again to make sure I was not missing anything.

One of the main concerns I raised was about the claims about how general the empirical results are claimed to be, when at the end they are an ablation over OTIL. Well, this was somewhat addressed in the Abstract and Introduction, but surprisingly, not everywhere in the paper. For example, right now it reads:

> *We study these different design choices both from a theoretical perspective and via a thorough ablation study. We characterize unexpected behaviors that can positively or negatively impact recovering the expert policy. Based on these insights, we propose OTIL, an intuitive OT method for IL that significantly improves upon previous methods w.r.t. sample efficiency (number of interactions with the environment), simplicity, generalizability (to pixel-based learning) and performance.*

This is very misleading. It is said: “Based on these insights, we propose OTIL”, but it is the other way around, by definition, no? OTIL was designed and then the ablation was performed? How can an ablation be made out of an algorithm that doesn’t exist? The direction of causality here seems flipped, but it changes the tone of the contributions and how the work was performed.

More than that, in Section 5 we have:

> *1. What are important design choices for imitation learning approaches leveraging optimal transport? OT cost – see Figure 3, state-based normalization – see Figure 4, and encoding for OT in the pixel case – see Figure 5, OT solver – see Figure 6, OT aggregation – see Figure 7, and squashing – see Figure 8.*

But again, this is misleading, because this is not about the design choices for imitation learning approaches leveraging optimal transport, but OTIL. The upsetting part is that this is a comment I raised and then I had to verify that it was not properly addressed, and I did read the whole paper again to see how the feedback to the authors was reflected in the new version. I don’t think it is reasonable to expect all that back and forth with the reviewers.

Other points I raised were ignored. For example, it reads *In all Deepmind control experiments, we run 10 seeds under each configuration and average results with a 90% confidence interval.* I explicitly asked how confidence intervals were computed, this was a requested change, and it was ignored.

The same applies to the empirical evaluation. Some claims were softened, as I mentioned above, but overall, I still don’t have the details I asked for. How were parameters swept, for example?  See my original review for more on that. I don’t think a paper should be published if there are not enough details to replicate the empirical methodology (ensuring the empirical methodology was sound) – I don’t think this is about code, it is about understanding the process for running experiments. Without these details, I cannot even judge the paper. I’ll say it again, just as an example: when a component is swapped by another, as in Figure 3, were parameters swept again? It wouldn’t be surprising that OTIL, which certainly uses the best hyperparameters found, ends up being better than a different version, with the Euclidean distance, for example, mainly if the hyperparameters for the latter were not tuned.

I genuinely didn’t want to be the reviewer that comes back in the rebuttal and asks for more stuff that they didn’t ask the first time. At this point, I’m sure this is not what I’m doing in this message, but just restressing the issues I raised in my first review. Hopefully one can understand that I’m not pleased to have spent another couple of hours evaluating a new version of the paper that didn’t address my initial comments.

---

> ### Author Response · Authors · 2022-08-04
> **Follow-up answer post-rebuttal**
>
> Thank you for thoroughly engaging in this reviewing process. We completely agree that our experimental results are ultimately ablations over OTIL in the domains we evaluate it on and upon re-reading the paper, understand how it is misleading for us to present them as more general principles. We are fully committed to updating these parts of the paper and have posted a significant new version of our paper changing our presentation here and toning down our claims. Can you please let us know if this new revision addresses your concerns and if there is anything else you would like for us to update? We apologize for not fully addressing these important issues in our first response.
>
> We have also updated the claim of a “unifying perspective” to simply saying that we summarize the key component of related work. While we have found the acronym “OTIL” appealing for our combination of these components, we also do not want to overclaim on the naming here too and are open to re-naming the method to make it clear that we are building on all of the other work in the OT-based imitation learning space, would you prefer a different name?
>
> > Other points I raised were ignored. For example, it reads In all Deepmind control experiments, we run 10 seeds under each configuration and average results with a 90% confidence interval. I explicitly asked how confidence intervals were computed, this was a requested change, and it was ignored.
>
> We’re sorry for missing the response to these details and are also committed to clarifying them so that our methodologies and ablations are as understandable and reproducible as possible. In the confidence intervals, we use [seaborn’s lineplot](https://seaborn.pydata.org/generated/seaborn.lineplot.html#seaborn-lineplot) to compute the 90% confidence interval over the average reward obtained from each seed with all of the other hyper-parameters fixed.
>
> > The same applies to the empirical evaluation. Some claims were softened, as I mentioned above, but overall, I still don’t have the details I asked for. How were parameters swept, for example? See my original review for more on that. I don’t think a paper should be published if there are not enough details to replicate the empirical methodology (ensuring the empirical methodology was sound) – I don’t think this is about code, it is about understanding the process for running experiments. Without these details, I cannot even judge the paper. I’ll say it again, just as an example: when a component is swapped by another, as in Figure 3, were parameters swept again? It wouldn’t be surprising that OTIL, which certainly uses the best hyperparameters found, ends up being better than a different version, with the Euclidean distance, for example, mainly if the hyperparameters for the latter were not tuned.
>
> We again apologize for missing this detail on this ablation. We indeed did not re-tune all of the other hyper-parameters when we ablated the cost in Figure 3 and agree that it could be possible for a method to work well with Euclidean costs if properly tuned even though the cosine cost appeared to work better for us.
>
>
> We hope that you will agree that the experimental details and demonstrations behind our main results in Figure 9 and 10 are fully explained, sound, and reproducible. To the best of our knowledge, these results are great baselines for the pixel-based IL community to have and are worth disseminating.
>
> In our revised version now, we have re-structured the paper to present these concrete results first and then showing the other ablations afterwards to help show what happens when we tweak some of the individual components. We believe that these are still interesting in the current form to give some intuition of stability, but we are also open to removing parts of them too as in some ways they are not crucial to the primary contribution of our paper. Please let us know if you have any suggestions here!

---

> > ### Author Response · Authors · 2022-08-11
> > **Follow-up answer post-rebuttal**
> >
> > Dear reviewer vqB1, we would greatly appreciate it if you could let us know if this clarified your concerns, and if you have any remaining questions. We look forward to addressing any remaining concerns before the end of the discussion period. Thank you very much for your time.

---

> > > ### Comment · Reviewer_vqB1 · 2022-08-11
> > > **On the confidence intervals**
> > >
> > > I didn't have the chance to look at the whole paper again. I'll get back to the authors when I do so. In the meantime, the answer about how the confidence interval was computed was not an answer. I want to know the procedure used, not what function was called. This particular function behaves differently depending on the parameters it receives.

---

### Comment · Reviewer_vqB1 · 2022-08-12
**Final recommendation**

I just recommended rejecting the paper. Given that this was my third pass over the paper, I decided to write another public review to justify my decision and to maybe help the authors in a future submission.

First and foremost, this was a frustrating process because, as I said in my previous comments, I felt my comments had not been addressed properly. When I read the paper again, I was hoping that this wouldn’t be an issue, but somehow it still is. I recognize the paper was changed more substantially than before, but still has issues. What is now Section 3.1 refers to propositions that only show up in Section 3.2, as if they had been already presented. The paper still uses language such as “We hence recommend leveraging a mean over the top-K closest” when we had agreed that the experiments are not enough to justify any recommendation, they are nothing more than an ablation. Much more important than that, literally, in Section 4, the first thing that is said is that the paper answers the question “What are important design choices for imitation learning approaches leveraging optimal transport?”, and the ablation studies are referred to as the answers. As I wrote more than once before, this is extremely misleading. Similar claims are made in the conclusion “extensive study of design choices through theory and empirical ablations”.

So, there’s the frustration part that makes me wonder how many times we would have to iterate over this paper to make it streamlined (it should have been only once in my opinion). But trying to leave that aside, the basis of my recommendation is the fact that out of the four questions listed in Section 4, the first one was definitely not answered, it is not clear to me why the third is relevant (I did ask that before as well), and the fourth is not meaningful: a method outperforming others cannot be seen as a big contribution, not in toy domains such as these.

To say it one more time: ablation studies are very valuable, as long as they are done carefully. OTIL was designed to use the components it starts with, every ablation, without a proper hyperparameter tuning, is very likely to perform worse (the used parameter was chosen for the other method); not to mention that the original approach also interacts well with the other components. I praise the authors for being open about the empirical methodology and I appreciate the honesty in Section 4.5. However, even though this was discussed at length, we still read things like “our approach performs best”, “we achieve best performance”. As acknowledged in the paper, “it could be possible for a method to work well with some of the less effective components if properly tuned”. Thus, why are any claims made in light of that relevant? Also, why call them “less effective components”?

Thus, to answer the questions listed in the TMLR guidelines for reviewers, the claims made in the submission **are not** supported by accurate, convincing and clear evidence.

There are other minor things that shouldn’t be there at this point, such as missing references (Section 4.1), claims such as “All proofs can be found in Appendix E” when all but one proof is in Section 3.2, and how my question about the confidence interval was not responded (although this one is not minor) — I don’t care if matplotlib, scipy, or seaborn was used to compute it, I want to know if assumptions about normality were made, if bootstrapping was used, etc.

---

### Decision · Action_Editors · 2022-08-19

**Recommendation:** Reject

**Comment:**

I would like to thank the authors for submitting their work to TMLR, and graciously engaging with the reviewers throughout the submission process. I believe that the reviews and the discussion that followed the initial rebuttal resulted in steady improvements to the paper, but unfortunately not to a degree to fully meet TMLR's requirements for publication. Unfortunately I don't think we reached a point where minor revisions are sufficient to capture the reviewers outstanding feedback (who won't have an opportunity to respond), and I strongly encourage the authors to further iterate and incorporate reviewer vqB1's and mU4T's feedback into the paper, and resubmit to TMLR. I would be happy to serve as AE for a second submission.

The evidence presented in the submission was not deemed convincing enough to support some the claims and general recommendations present in the paper. I suggest that either claims are _consistently_ adjusted in the paper, or preferably some of the relative evaluation methodologies - and some details of their descriptions - are to be strengthened in support of the claims. In particular, despite the reformatting of the order of results in the paper, one reviewer found the framing around the ablation study misleading, and the ablation methodology weak. One suggestion would be the retuning of hyper-parameters after each component of the algorithm is modified, and ensuring that details of how results are presented are reported explicitly, e.g. how confidence interval were computed.

Reviewer vqB1's concerns around using an ablation to justify general design recommendations are worsened by the observations from reviewer mU4T, which looked into the presented theory as guiding principle for the formulation of OTIL, but didn't find the contributions informative enough to guide the practice, either because they were a bit simplistic or detached from the practice. Reframing the theory contribution as an analysis section rather than a subsection used to design the OTIL algorithm did help with softening the claims around the theory contributions, but implicitly did set higher expectation on the design & ablation component of the paper.

Given how encouraging the empirical results demonstrated in the paper are, I hope the authors will follow up on the feedback, and decide to resubmit at TMLR soon.